

# The tropospheric response to zonally asymmetric momentum torques: implications for the downward response to wave reflection and SSW events

Wuhan Ning[1], Chaim I. Garfinkel[1], Judah Cohen[2,3], Ian P. White[4], and Jian Rao[5]

[1]Fredy & Nadine Herrmann Institute of Earth Sciences, The Hebrew University of Jerusalem, Israel

[2]Atmospheric and Environmental Research, Janus Research Group LLC, Lexington, Massachusetts 02421, USA

[3]Department of Civil and Environmental Engineering, Massachusetts Institute of Technology, Cambridge, Massachusetts, 02139, USA

[4]Bureau of Meteorology, Melbourne, Victoria, Australia

[5]State Key Laboratory of Environment Characteristics and Effects for Near-space, Nanjing University of Information Science and Technology, Nanjing 210044, China

**Correspondence:** Chaim I. Garfinkel (chaim.garfinkel@mail.huji.ac.il)

**Abstract.** The role of zonal structure in the stratospheric polar vortex for the surface response to weak vortex states is isolated using an intermediate-complexity moist general circulation model. Zonally asymmetric wave-1 momentum torques with varying longitudinal phases are transiently imposed in the stratosphere to induce stratospheric sudden warmings (SSWs) and wave reflection events, and the subsequent tropospheric and surface response is diagnosed. The response in these torque-induced
SSWs is compared to 48 spontaneous SSWs in the control experiments. Wave-1 forcings with opposite phases (centered at 90E vs. 270E) induce contrasting influences in the stratosphere and troposphere, including oppositely shifted polar vortex and opposite structures of zonal wind anomalies. Notably, downward wave propagation predominantly occurs over North America in the phase-90 ensemble, while primarily over North Eurasia in the phase-270 ensemble. These differences extend to surface responses: the phase-90 ensemble features pronounced cooling over Alaska and eastern Eurasia, along with enhanced rainfall
concentrated over the North Pacific and the North Atlantic, extending to northwest Europe. In contrast, the phase-270 ensemble exhibits significant cooling over central North America and North Eurasia, accompanied by enhanced rainfall over the North Pacific and the North Atlantic, stretching into subtropical Eurasia. By analyzing the mass streamfunction of the divergent component of the meridional wind, we observe opposite-signed zonal dipole patterns between the stratosphere and free troposphere, which further elucidates the pathway of stratosphere-troposphere coupling associated with SSWs and downward wave
propagation events. In the lower troposphere, the meridional mass streamfunction is linked to the surface cooling and warming responses to both SSWs and wave reflection events, as reported in previous studies, by illustrating the meridional advection of cold and warm air masses. Overall, this study indicates that the observed surface response following SSWs and stratospheric wave reflection events is a genuine signal arising from, and causally forced by, stratospheric perturbations.



## 1 Introduction

Stratospheric sudden warmings (SSWs) are the most extreme dynamical disturbance in the stratosphere, characterized by rapid and dramatic variability in both temperature and wind fields within just a few days over the polar regions (Baldwin et al., 2021). Historically, SSWs are observed almost exclusively during midwinter in the Northern Hemisphere (NH) (Butler et al., 2015; Charlton and Polvani, 2007; Cohen and Jones, 2011; Rao et al., 2021b). In contrast, the Southern Hemisphere (SH) has experienced only two SSWs, the first occurring in September 2002 and the second, a minor SSW, occurring in

late September 2019 (Rao et al., 2020b; Thompson et al., 2005). This distinct hemispheric asymmetry is attributed to the much weaker planetary-scale wave activity in the SH, which results from a smoother land–sea contrast and less pronounced orographic features in the SH compared to the NH (Liu et al., 2025; Scaife and James, 2000). During the forcing stage of SSWs, zonal mean zonal winds at 60°N and 10 hPa transition from strong westerlies to weak easterlies, accompanied by rapid increases in polar temperature by several tens of degrees (Huang et al., 2022; White et al., 2021). Notably, these disturbances

can propagate downward into the troposphere, impacting the surface weather through stratosphere-troposphere coupling and triggering cold air outbreaks across Eurasia and North America (Cohen et al., 2022; Domeisen et al., 2020; Garfinkel et al., 2017).

SSWs are often categorized as either displacement or split events based on the geometry of the polar vortex disruption. In displacement events, the polar vortex shifts away from the pole, whereas split events involve the vortex breaking into two

separate vortices (Butler et al., 2017; Seviour et al., 2013). These different morphologies arise due to differences in both the tropospheric precursors and stratospheric initial conditions of SSWs (Cohen and Jones, 2011; Garfinkel et al., 2010), and also from nonlinear interactions within the stratosphere (Rupp et al., 2025). Some studies suggest that wave-1 forcing contributes to both split and displacement events, while wave-2 forcing is more strongly associated with split events in particular (Bancala et al., 2012; White et al., 2019).

Studies using reanalysis data have suggested that these differences in SSW morphology may lead to distinct impacts on surface weather and climate (Hall et al., 2021; Mitchell et al., 2013; Seviour et al., 2013). For example, Mitchell et al. (2013) reported that observed split events intend to produce stronger and more persistent near-surface anomalies than displacement events, while Hall et al. (2021) focused on the weeks immediately after the SSW and showed that significant surface cooling is typically observed over northwest Europe and north Eurasia during split events, and enhanced rainfall is often observed

over northwest Europe in displacement events. While these studies provide valuable insights into the surface manifestations in different SSW types, they primarily rely on statistical composites of historical displacement and split events, and they do not establish a causal mechanism for the observed surface anomalies. Earlier modeling work on the difference in impacts did not show a clear difference, assuming one controls for the magnitude of the underlying SSW (Maycock and Hitchcock, 2015), nor do large hindcasts of SSW events from subseasonal forecast models (Rao et al., 2020a).

To more cleanly distinguish the near-surface impacts associated with displacement and split events, White et al. (2021) employed an idealized moist model, imposing wave-1 and wave-2 switch-on/off heatings in the stratosphere to mimic the sudden nature of an SSW. Their analysis revealed substantial differences between displacement and split events in the control and




wave-forced experiments, particularly at short time lags following the onset date. However, at longer time lags, the differences between wave-1 forced displacement events and wave-2 forced split events become minimal. The model simulations presented by White et al. (2021) have two key limitations, however. First, White et al. (2021) explored only a limited number of longitudinal phases for the imposed heating forcing, which may be insufficient to fully capture the variability and distinguish features between displacement and split events. More critically, the heating-based forcing scheme utilized by White et al. (2021) induces a qualitatively opposite effect on the meridional circulation that occurs in response to an imposed heating, which is known as the "Eliassen adjustment" (Eliassen, 1951), leading to a reversed Brewer-Dobson circulation (BDC), and thereby failing to accurately reproduce the initial dynamical response during the forcing stage of SSW events (White et al., 2022).

To address this limitation, White et al. (2022) introduced an alternative approach involving an imposed easterly momentum torque to the stratosphere. This methodology generates a BDC response similar to that observed during naturally occurring SSWs, including poleward flow in the stratosphere, downwelling over the pole, and equatorward flow in the troposphere, consistent with the dynamics during spontaneously occurring SSWs. Note that this response is the "Eliassen adjustment" to an imposed momentum torque (Eliassen, 1951). However, the momentum torque as imposed in White et al. (2022) is zonally symmetric and, therefore, unable to capture the distinct effects of zonal asymmetries of forcing in the stratospheric vortex.

In the present study, we extend the framework of White et al. (2022) by applying a zonally asymmetric torque in the stratosphere. This approach enables a more comprehensive assessment of how zonal asymmetries of wave forcing influence surface responses following SSWs. Note that this asymmetric forcing is a transient mechanical forcing, which better simulates the sudden and transient nature of planetary-wave forcing that drives an SSW in the first place (White et al., 2022).

SSWs are not the only extratropical stratospheric phenomena that have been linked to a pronounced surface impact. On occasion, vertically propagating waves are reflected downward from the stratosphere back into the troposphere instead of being fully absorbed by the stratosphere, and when such a wave reflection event occurs, cold extremes in North America often follow. For example, while no SSW event was observed in the winter of 2013/2014, an exceptionally severe cold air outbreak still occurred across North America shortly after a wave reflection event (Cohen et al., 2022). Downward wave reflection can reinforce a strong blocking high-pressure system over Alaska and enhance a downstream trough over central North America, ultimately driving the extreme surface cooling (Agel et al., 2025; Cohen et al., 2021, 2022; Kretschmer et al., 2018a, b; Shen et al., 2023).

The causality of associating this surface impact with the stratospheric reflection event is not straightforward, however. For instance, Matthias and Kretschmer (2020) reported that the tropospheric response to wave reflection events tends to be nearly instantaneous and relatively short-lived, typically lasting from several days to about two weeks. Furthermore, some studies have shown that cold anomalies over North America tend to reach peak intensity during and immediately following such wave reflection episodes (Kretschmer et al., 2018a). Hence, it is unclear if the surface cold is due to the reflection or rather due to a precursor pattern in the troposphere that also happened to have contributed to a wave pulse that was subsequently reflected downward. While previous work has demonstrated a causal influence from SSWs to the near-surface by imposing or nudging an SSW in a model (Hitchcock and Simpson, 2014; White et al., 2020, 2022), we are unaware of similar work for reflection events. Therefore, in the present study, we aim to isolate the downward impact from the wave reflection events that often occur





following SSW events (e.g., the January 2021 SSW, Cohen et al., 2021) or in near-miss SSW events (Cohen et al., 2022), and then elucidate the dynamical mechanisms by which they influence near-surface weather, particularly cold anomalies over

North America.

We introduce the model setup and our control and wave-forced experiments in Section 2. Section 3 describes the diagnostic tools for analysis. Our main results are shown in Section 4, and a discussion and summary are presented in Section 5.

## 2 Model and experimental setup

To fully understand the complexities of the climate system and isolate key processes, it is essential to explore the dynamic

changes in idealized nonlinear climate systems by systematically adding or subtracting key processes, which contribute to the complexity of the global atmosphere (Hoskins, 1983). In this study, we use the Model of an idealized Moist Atmosphere (MiMA), recently developed by Jucker and Gerber (2017), Garfinkel et al. (2020a), and Garfinkel et al. (2020c). MiMA is an atmospheric general circulation model (GCM) of intermediate complexity, bridging the gap between the more idealized dry GCMs and fully comprehensive models of the real atmosphere.

### 2.1 Model of an idealized Moist Atmosphere (MiMA)

Building on the aquaplanet models of Frierson et al. (2006) and Merlis et al. (2013), Jucker and Gerber (2017) introduced a full radiative transfer scheme - the GCM version of the Rapid Radiative Transfer Model (RRTM) (Mlawer et al., 1997) - and more realistic surface forcing to more accurately represent a range of physical processes. The most important of these physical processes in this study is a realistic treatment of the interactions of shortwave radiation with stratospheric ozone, which

allows MiMA to better represent the stratospheric circulation. MiMA also incorporates a physically consistent representation of moisture transport and latent heat release, within a parameterized convection scheme and a resolved transport scheme (Betts, 1986). Additionally, it features an idealized boundary layer scheme based on Monin-Obukhov similarity theory and a slab ocean. For further details on the model configuration, please see Jucker and Gerber (2017) and Garfinkel et al. (2020a).

MiMA incorporates an interactive parameterization scheme for gravity wave deposition (Alexander and Dunkerton, 1999).

Following Garfinkel et al. (2020c), gravity wave momentum flux, which would otherwise transit the model lid (and hence violate momentum conservation; Shepherd and Shaw, 2004), is deposited within the top three pressure levels. Anomalously intense upward fluxes of planetary waves from the troposphere play a role in the development of many NH SSWs (Cohen and Jones, 2011; Garfinkel et al., 2010; Polvani and Waugh, 2004; Rao et al., 2021a; White et al., 2019). In addition, stationary waves affect both the climate and weather in the troposphere over broad latitude bands (Simpson et al., 2016). Thus, it is vital to

simulate the stationary waves as realistically as possible to reproduce the SSWs in the present study. Following Garfinkel et al. (2020b), we modify the lower boundary conditions of MiMA to generate a stationary wave pattern that closely resembles those seen in CMIP models. This is achieved by employing three key forcing mechanisms of stationary waves: 1) topography, 2) ocean horizontal heat flux, and 3) land-sea contrast (Garfinkel et al., 2020a, b). Notably, there are differences in how the lower boundary is modified between the first version of MiMA constructed by Jucker and Gerber (2017) and the current configuration



used in this study. For more details on the construction of stationary waves, please refer to Garfinkel et al. (2020b). We also replace the annually averaged ozone input data with monthly climatological zonal-mean ozone input data, sourced from the preindustrial era CMIP5 forcing. This ozone forcing is also adopted by White et al. (2020) in their investigation of tropospheric responses to SSWs by using MiMA. All simulations are constructed at a horizontal resolution of triangular truncation 42 (T42; $2.8° \times 2.8°$), with 40 vertical levels extending up to $\sim 0.1 \, \mathrm{hPa}$.

We extend the framework of the model configuration described in White et al. (2022), with several modifications made to the control and, in particular, the forced experiments.

## 2.2 Control experiments and SSW identification

We produce an ensemble of nine control runs by introducing slight variations to the parameterization of gravity waves during the first year of the model integration. The control run with the median value of gravity wave drag is designated as CTRL. This
CTRL run also serves as the basis for a series of branch ensembles, as detailed in the following subsection. Each control run is integrated for 50 years (360-day year) after discarding an initial 19-year spin-up period (including the year in which the gravity wave settings differ), ensuring the mixed-layer ocean eventually reaches an equilibrium state for later analysis.

Following Charlton and Polvani (2007), we identify SSWs in our control runs but restrict their occurrence to the Northern Hemisphere winter months of January through March (JFM), excluding events in November and December. Because the branch
ensembles below use each midnight on the 1st January from the CTRL run as their initial conditions, it is reasonable to consider SSWs only occurring within the JFM period across all control runs. Therefore, the definition of SSWs, adapted from Charlton and Polvani (2007) with minor modifications to suit the present study, includes the following three criteria: 1) Zonal-mean zonal wind at $60°$N and 10 hPa must reverse from westerly to easterly during the JFM period, and the date of this reversal is defined as the onset or central date of the SSW. 2) After the reversal, the zonal-mean zonal wind must remain easterly for at
least 10 consecutive days before April 30, to exclude final warming events. 3) Two consecutive SSWs are considered separate events if their onset dates are at least 20 consecutive days apart within the JFM period. In all, 48 SSW events are identified across all nine control runs during the JFM period.

## 2.3 Branch experiments

As aforementioned, we branch off from the 50-year CTRL simulation at 00:00 on the 1st January of every year, and impose
a momentum torque in the stratosphere. 50 ensemble members are created. We build on the momentum torque used in White et al. (2022), with the key new feature being that we allow for zonal structure. This momentum torque is introduced into the model's zonal momentum budget as an external forcing and is formulated as follows:

$$F(t,\varphi,\lambda,p) = \tau(t)\Phi(\varphi)\Lambda(\lambda)H(p) \tag{1}$$





where

$$\tau(t) = \begin{cases} 1, & \text{if } 0 < t - t_0 \le N_d \text{ days} \\ 0, & \text{otherwise} \end{cases} \tag{2}$$

$$\Phi(\varphi) = M_S \sin\left(\pi \frac{\varphi - \varphi_L}{\varphi_H - \varphi_L}\right) \tag{3}$$

$$\Lambda(\lambda) = 1 + \Theta_A \sin\left(k\pi\left(\lambda - \lambda_0\right)/180\right) \tag{4}$$

and

$$H(p) = \begin{cases} \frac{p - p_b}{p_t - p_b}, & \text{if } p_t < p < p_b \\ 1, & \text{if } p \le p_t \\ 0, & p > p_b \end{cases} \tag{5}$$

In the above equations, $t$, $\varphi$, $\lambda$, and $p$ represent model variables corresponding to time, latitude, longitude, and pressure, respectively. All other parameters are tunable to define the characteristics of the imposed momentum torques. Specifically, $N_d$ denotes the prescribed duration of the perturbation, with the applied forcing activated at the reference time ($t_0$, the midnight on 1st January). The parameter $M_S$ represents the amplitude of the zonally symmetric component of the momentum torque. The latitude bounds $\varphi_L$ and $\varphi_H$ define the region where the forcing is applied. Similarly, $p_b$ and $p_t$ specify the lower and upper pressure levels, where the torque decreases linearly with pressure. $\Theta_A$, $\lambda_0$, and $k$ present the scaling factor, the longitudinal phase, and the zonal wave number of the imposed asymmetric momentum torque, respectively. This $\Theta_A \sin\left(k\pi\left(\lambda - \lambda_0\right)/180\right)$ term is the key new term introduced in this paper, which varies sinusoidally with longitude.

In their study, White et al. (2022) calculated the divergence of Eliassen–Palm flux (EPFD) to reveal the spatial-temporal structure of EPFD during the forcing stage for 70 SSWs in control runs. Then they utilized the parameter settings mentioned above to mimic the EPFD signature by imposing a symmetric momentum torque on the zonal momentum budget. For more details on the spatial-temporal variability of symmetric momentum torque, please see Fig.3 of White et al. (2022). In this study, we also conduct the zonally-symmetric branch ensemble using the same configuration of momentum torque as designed by White et al. (2022), which are as follows: the perturbation duration $N_d$ is set to be 12 days, and zonally symmetric momentum torque $M_S$ is -15 m s$^{-1}$ day$^{-1}$. The latitude bounds are $\varphi_L = 40°$N and $\varphi_H = 90°$N, centering the forcing around 65°N. The forcing is at full strength above $p_t = 60$ hPa, and decreases linearly to zero at $p_b = 100$ hPa.

In this study, the scaling factor for the asymmetric forcing component, $\Theta_A$, is set to 4 to ensure that the SSW events have a strong zonally asymmetric component. We specifically focus on two perturbation experiments in the present study, phase-90 ensemble ($k = 1$, $\Theta_A = 4$, $\lambda_0 = 90°$) and phase-270 ensemble ($k = 1$, $\Theta_A = 4$, $\lambda_0 = 270°$), which result in a similar response of surface temperature compared to observational results based on clustering analysis (Agel et al., 2025; Kretschmer et al., 2018a). Ongoing work is aimed at analyzing the surface response for other choices of $k$ and $\lambda_0$.

Note that all branch ensembles are integrated for 120 days to allow sufficient time for the stratosphere to recover following the SSW event, and all consist of 50 ensemble members. The anomalies in branch ensembles are compared to the climatology



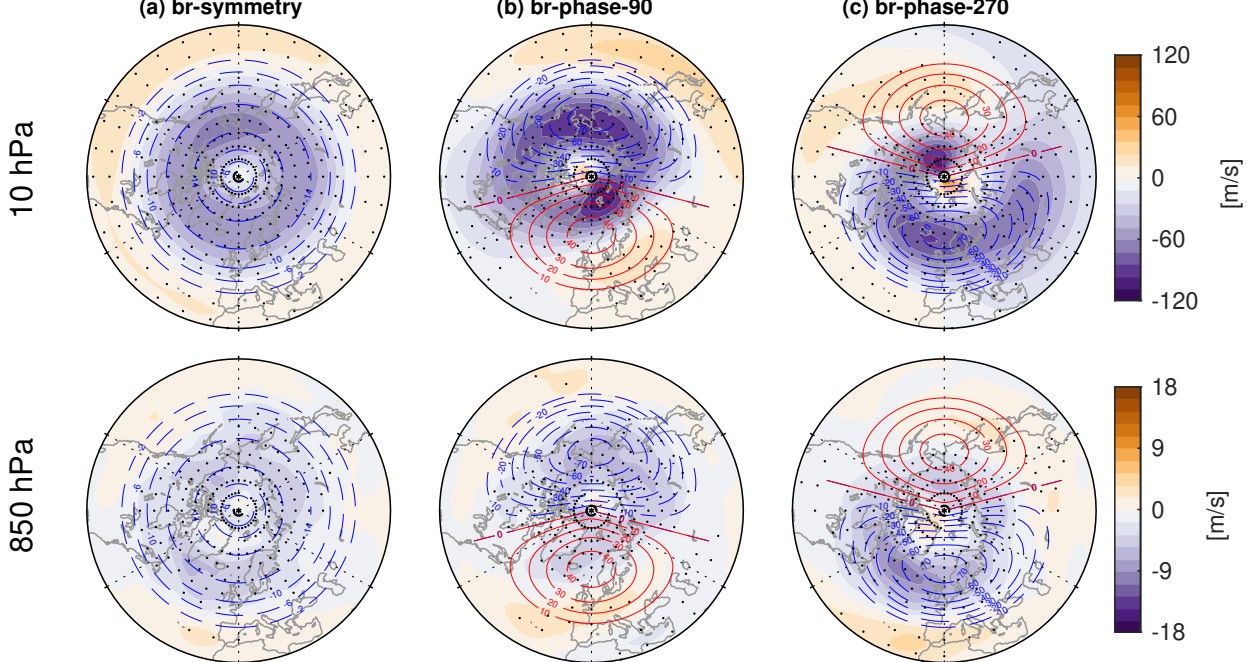

**Figure 1.** Zonal wind anomalies (shading; m s$^{-1}$) at 10 hPa (upper row) and 850 hPa (bottom row) during days 6 to 12 for (a) symmetric branch ensemble, (b) phase-90 branch ensemble, and (c) phase-270 branch ensemble. The blue dashed (red solid) contour indicates the applied negative (positive) wave forcing (contours; m s$^{-1}$ day$^{-1}$) at 10 hPa. The contour interval for wave forcing is 4 m s$^{-1}$ day$^{-1}$ in (a) and 10 m s$^{-1}$ day$^{-1}$ in (b) and (c). Stippling indicates anomalies that are statistically significant at the 95% level based on Student's $t$ test. Note that the forcing is applied only above 100 hPa but displayed at 850 hPa for visual comparison.

of the CTRL run, while the anomalies in control runs are compared to the climatology of all nine control runs. Day 0 in control runs indicates the onset day of SSWs, while day 0 in branch ensembles is the 1st January of every year.

Figure 1 shows the spatial distributions of the imposed forcing (contours) at 10 hPa for three branch ensembles. In the zonally symmetric torque ensemble (Figure 1a), the forcing peaks at -15 m s$^{-1}$ day$^{-1}$ along the 65°N latitude circle and weakens both equatorward and poleward until zero. In contrast, in the two asymmetric ensembles (Figure 1b,c), the wave-1 forcing varies sinusoidally along the zonal circle, with a maximum value of 45 m s$^{-1}$ day$^{-1}$ and a minimum value of -75 m s$^{-1}$ day$^{-1}$ at 65°N. By construction, the spatial distribution of wave-1 forcing in the phase-90 and phase-270 ensembles is 180° out of

phase. Note that all three branch ensembles apply an identical net wave forcing of -64.5 m s$^{-1}$ integrated over the specified latitude-pressure region during the perturbation period ($N_d = 12$ days), which is the same as the net forcing adopted by White et al. (2022).



## 3 Diagnostic Tools

### 3.1 Mass streamfunction

Given the nature of the imposed forcing, a meridional circulation is expected to develop akin to the meridional circulation that develops in response to a zonally symmetric heating or torque (Eliassen, 1951; White et al., 2021, 2022), but with a strong zonally asymmetric component if $\Theta_A$ is nonzero. To diagnose this meridional circulation, we use the psi vector method originally developed by Keyser et al. (1989). They developed this method to partition the three-dimensional circulation into orthogonal two-dimensional circulations, to describe the vertical motion and horizontal irrotational flow within mid-latitude

frontal systems. Building on this psi vector, Schwendike et al. (2014) proposed a new version of the psi vector, which is formulated in spherical coordinates and is used to analyze the local Hadley circulation and local Walker circulation in the tropics. We use this psi vector method of Schwendike et al. (2014) as implemented by Raiter et al. (2024) to investigate the mass streamfuction by the divergent component of wind induced by the stratospheric torque. The streamfunction in pressure coordinates can be specified as:

$$\Psi_\lambda = \frac{1}{g} \int_0^p u_{\mathrm{div}} dp \quad \text{and} \quad \Psi_\phi = \frac{1}{g} \int_0^p v_{\mathrm{div}} dp \qquad (6)$$

where $g$ is the gravitational acceleration, $u_{\mathrm{div}}$ and $v_{\mathrm{div}}$ represent the divergent component of zonal wind and meridional wind, and $\Psi_\lambda$ and $\Psi_\phi$ denote the zonal and meridional mass streamfunction, respectively. See Raiter et al. (2024) and Schwendike et al. (2014) for further details. The meridional mass streamfunction $\Psi_\phi$ will be the focus of our analysis in the following section.

### 3.2 Wave activity flux

The Eliassen-Palm (EP) flux can successfully capture the propagation of Rossby waves and assess their interaction with zonal-mean flow, which has been proven to be a significant analysis tool (Andrews and Mcintyre, 1976; Edmon Jr et al., 1980). However, the EP flux is limited to representing the characteristics of waves in the meridional-vertical plane. To analyze the three-dimensional propagation of Rossby waves, we calculate the wave activity flux (WAF) on the sphere ($F_s$), following the

formulation derived by Plumb (1985) under the quasi-geostrophic assumption:

$$\boldsymbol{F_s} = \begin{pmatrix} F_x \\ F_y \\ F_z \end{pmatrix} = p\cos\phi \begin{pmatrix} v'^2 - \frac{1}{2\Omega a \sin 2\phi} \frac{\partial(v'\Phi')}{\partial\lambda} \\ -u'v' + \frac{1}{2\Omega a \sin 2\phi} \frac{\partial(u'\Phi')}{\partial\lambda} \\ \frac{2\Omega\sin\phi}{S}\left[v'T' - \frac{1}{2\Omega a \sin 2\phi} \frac{\partial(T'\Phi')}{\partial\lambda}\right] \end{pmatrix} \qquad (7)$$

where $F_x$, $F_y$, and $F_z$ represent the zonal, meridional, and vertical components, respectively, $p = \frac{\text{pressure}}{p_0}$ while $p_0 = 1000$ hPa, $\lambda$ and $\phi$ denote the longitude and latitude, $\Omega$ is the rotation angular velocity of the Earth, $a$ is the mean radius of the Earth, $u$ and $v$ refer to the zonal and meridional wind, $T$ is the air temperature, $\Phi$ is geopotential, and $S$ is the static stability parameter:






$$S = \frac{\partial \hat{T}}{\partial z} + \frac{k\hat{T}}{H} \qquad (8)$$

where $\hat{T}$ indicates areal-averaged temperature over the pole cap north of 20°N, $k$ is the ratio of the gas constant $R$ to the specific heat at constant pressure $C_p$, $H$ is the constant scale height (7 km), and $z = -H\ln(p)$ represents the log-pressure vertical coordinate. In this study, $F_s$ is calculated only for stationary waves with zero phase speed, and it is filtered to retain only the first three zonal wave number components, as waves 1-3 dominate the wave activity flux in the Northern Hemisphere
polar stratosphere (Cohen et al., 2022).

## 4   Results

### 4.1   Stratospheric and tropospheric responses

We begin by examining the impact of the imposed momentum torques on zonal wind. Figure 1 illustrates zonal wind
anomalies (shading) and applied forcing (contours) during days 6 to 12 (the latter half of the period when the forcing is switched on) across the three branch ensembles. As discussed in Section 2.3, momentum torques are imposed only in the stratosphere. To aid visualization, they are also included in the bottom panels in Figure 1 that show the wind response at 850 hPa. As expected, the momentum torque decelerates zonal wind, with the deceleration concentrated in the sector with the zonally asymmetric negative forcing. In the stratosphere, zonal wind weakening is observed from 40° N to the pole in the
symmetry ensemble (Figure 1a). This zonally symmetric response presents a uniform deceleration of the polar vortex under the impact of zonally symmetric forcing. However, in the phase-90 and phase-270 ensembles (Figure 1b,c), the zonal wind anomalies peak in opposite hemispheres due to the reversed phase of the imposed wave-1 forcing. Comparing the zonal wind at 850 hPa with that at 10 hPa reveals a strong resemblance in their spatial patterns across these branch ensembles.

To better investigate the response of the polar vortex, Figure 2 presents both the raw geopotential height field at 10 hPa and
the anomaly geopotential height field at 500 hPa during days 1 to 5 (shortly after the forcing is switched on). In the stratosphere, the polar vortex anomalies are positioned more above the center of the pole in the symmetry ensemble (Figure 2b). Nonetheless, there is a slight zonal structure in both the control SSWs and the symmetry ensemble: the daughter polar vortex is found over northern Europe, while the Alaskan High is strengthened; this configuration resembles observed displacement SSWs as well (Matthewman et al., 2009). This zonal structure of the polar vortex is substantially more pronounced in the phase-90 and
phase-270 ensembles (Figure 2c,d): the vortex exhibits a clear displacement away from the pole, but more importantly, the direction of displacement is opposite between these two experiments. These opposite shifts indicate that the opposite phase of imposed wave-1 forcing induces opposite modifications to the polar vortex, altering its central position and overall structure. In the troposphere, the symmetry ensemble exhibits an anomalous high largely located above the pole (Figure 2b). In contrast, in the phase-90 and phase-270 ensembles (Figure 2c,d), the anomalous high exhibits significant displacements to opposite
sectors, with the tropospheric anomaly directly underlaying the strongest anomalies in the mid-stratosphere. This suggests that





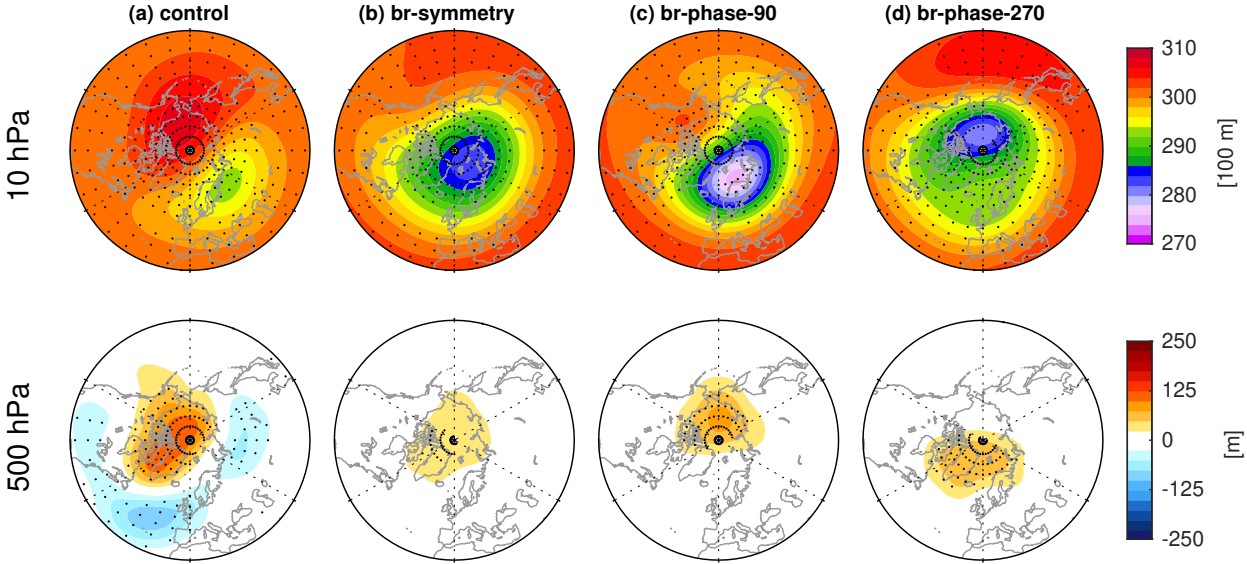

**Figure 2.** Geopotential height raw fields at 10 hPa (upper row) and anomalies at 500 hPa (bottom row) during days 1 to 5 for (a) control run, (b) symmetric branch ensemble, (c) phase-90 ensemble, and (d) phase-270 ensemble. Stippling indicates that anomalies at 500 hPa (bottom row) are statistically significant at the 95% level, while raw fields at 10 hPa (upper row) are statistically significantly different from the climatologies (not shown) at the 95% level, both based on Student's $t$ test.

the opposite phase of wave-1 forcing not only modifies the polar vortex in the stratosphere but also influences the tropospheric circulation through downward coupling. The relevant mechanisms will be discussed in section 4.4.

The temporal evolution of the stratospheric response is examined in Figure 3a and b, which presents the zonal mean zonal wind at $60°$N and 10 hPa and area- and pressure-averaged temperature anomalies between $50°$–$90°$N over 150–1 hPa. In
the control runs, the detected SSWs capture the reversal of the zonal mean zonal wind at $60°$N and 10 hPa, which suddenly transitions from westerlies to easterlies (Figure 3a), and warming of the polar stratosphere (Figure 3b). In the branch ensembles, during the forcing stage (days 1 to 12; the vertical magenta dashed line indicates day 12), SSWs are forced to occur by the imposed momentum torque, capturing the characteristics of naturally-occurring SSWs with reversal of the zonal mean zonal wind and warming of the polar stratosphere (Figure 3a,b). The SSWs in the control and branch ensembles are then followed by
a response in polar cap height at 51hPa and 500hPa (Figure 3c,d). It is primarily driven by a strengthened BDC accompanied by intensified downwelling over the pole (Baldwin et al., 2021). Following the peak phase, the polar cap gradually cools and returns to climatological temperatures and height over the subsequent four months. These anomalous stratospheric behaviors propagate downward and induce distinct impacts on tropospheric geopotential height patterns (Figure 3d).

Although the accumulated negative forcing is identical across the symmetry, phase-90, and phase-270 ensembles, significant
differences are evident in the tropospheric (and also stratospheric) responses, indicating that the longitudinal phase of the





**Figure 3.** Time series of (a) $\bar{u}$ (m s$^{-1}$) at $60°$N and 10 hPa, (b) $\bar{T}$ ($K$) area-averaged over $50° - 90°$N and pressure-weighted over 150 - 1 hPa, and polar cap geopotential height anomalies (m) area-averaged over $50° - 90°$N at (c) 51 hPa and (d) 500 hPa for the 48 control SSWs (black line) and for various branch ensembles (colored lines) from lag day -30 to 120. Note that there are 50 ensemble members for each branch ensemble. Lag day 0 indicates the onset of SSWs in the control run and initialization day in branch ensembles. The vertical magenta dashed line indicates day 12, the last day with imposed momentum torques in branch ensembles.



imposed wave-1 forcing plays a crucial role in shaping both stratospheric and tropospheric responses. How do these variations in stratospheric response translate into differences in surface conditions?

## 4.2 Surface responses

We begin by analyzing the temporal evolution of 2-m temperature anomalies (Figure 4). During days 1 to 12 (while the
stratospheric torque is still on), cooling and warming signals emerge but differ widely across the symmetry, phase-90, and phase-270 ensembles (Figure 4b,c,d). From day 13 onward, these anomalies intensify, suggesting a continued influence of imposed forcing on surface temperature patterns even after the forcing is turned off. In the control run and symmetry ensemble (Figure 4a,b), significant cooling occurs over North Eurasia and North America, while warming is observed over subtropical Eurasia and northeast Canada. This temperature response pattern resembles that found in reanalyses (Butler et al., 2017).
However, the cooling and warming patterns vary substantially in the phase-90 and phase-270 ensembles. Cooling is most pronounced over Alaska and east Eurasia in the phase-90 ensemble (Figure 4c), whereas in the phase-270 ensemble (Figure 4d), it is primarily concentrated over central North America and north Eurasia. These distinct cooling patterns are consistent with those found during clusters 4 and 5 in Kretschmer et al. (2018a), who apply hierarchical clustering analysis to daily-mean ERA-Interim data from 1979 to 2018. The warming patterns also differ between the two asymmetric ensembles. In the phase-
90 ensemble (Figure 4c), warming is primarily located over west Eurasia, whereas in the phase-270 ensemble (Figure 4d), it is more prominent over North America and subtropical Eurasia. The persistence and amplification of these anomalies after day 13 (after the forcing is switched off) indicate a causal coupling between the stratosphere and surface, as the stratosphere possesses a longer memory than the troposphere and can exert a prolonged impact on surface conditions (Baldwin et al., 2003). Moreover, the contrasting responses between phase-90 and phase-270 ensembles underscore the importance of the longitudinal
phase of asymmetric forcing in modulating regional surface temperature anomalies.

Next, the temporal evolution of precipitation anomalies is examined in Figure 5. During days 1 to 5, rainfall anomalies begin to emerge, and then these signals intensify significantly, exhibiting notable differences among the symmetry, phase-90, and phase-270 ensembles during days 6 to 19. After peaking during days 13-19, precipitation anomalies weaken rapidly, even as the temperature pattern persists in these days (Figure 4). This evolution suggests that the imposed forcing produces a temporary
but pronounced impact on regional precipitation, with rainfall anomalies gradually diminishing as the polar vortex recovers without the influence of momentum torques. By analyzing 24 SSWs that occurred in the 1979–2016 period using the ERA-Interim reanalysis, King et al. (2019) also reported a clear distinction between surface temperature and precipitation patterns: while a cooling signal can persist for up to two months following SSWs, rainfall tends to intensify in the first month and then it is drier in the second month over the Atlantic coast of northern Europe. Across all experiments, the strongest rainfall
anomalies are observed primarily over the North Pacific, extending into North America, and over the North Atlantic, stretching into subtropical Eurasia, which is consistent with the observational results of Butler et al. (2017). However, the distribution of anomalies differs widely between the phase-90 and phase-270 ensembles. Rainfall is particularly enhanced over subtropical Eurasia during days 13 to 19 in the phase-270 ensemble (Figure 5d), whereas in the phase-90 ensemble (Figure 5c), the enhancement of rainfall in Europe is weaker and located further north than for other experiments, while southeastern Europe



**Figure 4.** Temporal evolution of 2-m temperature anomalies during days 1 to 33 for (a) control run, (b) symmetric branch ensemble, (c) phase-90 branch ensemble, and (d) phase-270 branch ensemble. Stippling indicates statistically significant anomalies at the 95% level based on Student's $t$ test. Recall that the forcing in the branch ensembles is applied only from days 1 to 12 and is switched off from day 13 onward. Hence, we show the response from days 1 to 5 and from days 6 to 12 during the forcing stage, and weeks from day 13 onward during the recovery stage.







**Figure 5.** As in Figure 4 but for precipitation anomalies.





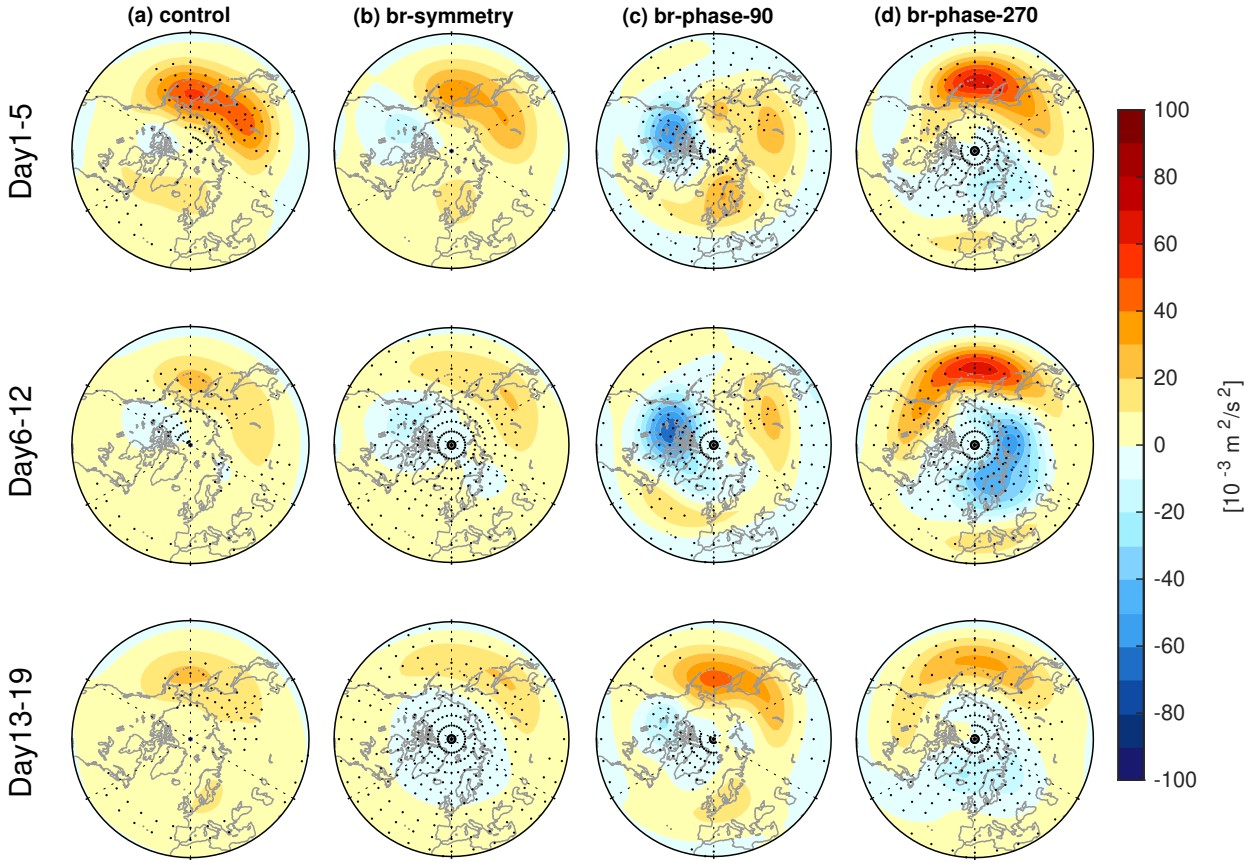

**Figure 6.** $F_z$ (raw field) at 93 hPa during days 1 to 19 for (a) control run, (b) symmetric branch ensemble, (c) phase-90 ensemble, and (d) phase-270 ensemble. Stippling indicates the raw values are statistically significantly different from the climatologies (not shown) at the 95% level based on Student's $t$ test.

dries. King et al. (2019) argued that the precipitation patterns align with sea level pressure anomalies before and after the SSW. While the present study will not delve into the mechanisms behind these precipitation anomalies, a detailed investigation will be presented in a subsequent paper.

### 4.3    Wave reflection and downward wave propagation

     Previous studies have argued that wave reflection events during weakened vortex states can lead to surface cooling over
North America and North Eurasia (Kretschmer et al., 2018a, b). To examine this relationship, we show the vertical component of Plumb flux $F_z$ at 93 hPa during days 1 to 19 in Figure 6. It shows the raw vertical Plumb flux field, not anomalies, and hence negative values of $F_z$ signify a wave reflection event accompanied by downward wave propagation, which can subsequently impact the tropospheric circulation and surface climate (Cohen et al., 2021, 2022). In the symmetry ensemble, downward wave propagation is more centrally located above the pole under the influence of symmetric forcing, but is generally weak.





Conversely, the phase-90 ensemble shows strong downward propagation localized over North America, which is then followed by local surface cooling even after the forcing is turned off (Figure 4c). A similar spatial pattern is also observed in the composite of cluster 4 events in Kretschmer et al. (2018a), who demonstrated that such events are associated with North American cold spells via reflected upward-propagating waves over eastern Siberia by using causal effect networks analysis. In the phase-270 ensemble, downward wave propagation is instead concentrated over North Eurasia, and is immediately followed

by pronounced surface cooling over Northern Eurasia (Figure 4d); this effect is consistent with the observational results of cluster 5 in Kretschmer et al. (2018a) and also cluster 5 in Cohen et al. (2021).

Note that wave reflection is influenced by the imposed momentum torque during the forcing stage from days 1 to 12. Even though the forcing is switched off during days 13 to 19, the surface temperature impacts appear strongest. This indicates that feedbacks are kicked off and influence the wave reflection even after the forcing is switched off. These results suggest

that the opposite phase of wave-1 forcing critically influences the positioning of the wave reflection event and, consequently, modulates regional surface temperature anomalies. A key question arises: through which dynamical pathway do these SSWs and downward wave propagation events ultimately affect surface conditions?

### 4.4 Downward coupling

Here, we delve deeper into the underlying mechanisms driving the surface cooling and warming due to SSWs and downward

wave propagation events. A key driver is the anomalous acceleration of the BDC, which plays a crucial role in stratosphere-troposphere coupling with downward motion over the pole, accompanied by enhanced poleward motion in the stratosphere. This acceleration of the BDC is due to the "Eliassen adjustment" which brings high angular momentum air from low latitudes to counteract the weakening of westerlies in high latitudes (Eliassen, 1951), and compensating equatorward motion in the troposphere (Baldwin et al., 2021; White et al., 2022). Specifically, White et al. (2022) found that the subsidence in the

troposphere over the pole and the subsequent near-surface northerly winds help induce the near-surface jet response in the first few days and weeks after the SSW onset before synoptic eddy feedbacks kick in.

To examine this effect, we calculate the mass streamfunction by pressure-integrating the divergent component of meridional wind over 34-10 hPa in Figure 7, and over 700-321 hPa in Figure 8 (see section 3.1). In the stratosphere (Figure 7), significant poleward motion is observed across all experiments. Under the influence of symmetric forcing, this poleward motion is broadly

distributed over the circumpolar region in the symmetry ensemble (Figure 7b). In contrast, the phase-90 ensemble exhibits poleward motion primarily over the eastern hemisphere during days 1 to 5 (Figure 7c), whereas in the phase-270 ensemble, it shifts to the western hemisphere (Figure 7d). This contrasting circulation pattern underscores how wave-1 forcing with opposite phase induces distinct and spatially opposing effects in the stratosphere.

A similar but opposite-signed circulation pattern emerges in the free troposphere (Figure 8), where equatorward motion is

significantly intensified across all experiments. Notably, in the phase-90 and phase-270 ensembles (Figure 8c,d), the tropospheric circulation exhibits a zonal dipole structure that is spatially aligned with the corresponding stratospheric circulation but with an opposite sign. Namely, tropospheric northerlies are strongest over Alaska in the phase-270 ensemble and over western Eurasia in the phase-90 ensemble even after the forcing has ended (Figure 8c,d). This correspondence between the streamfunc-





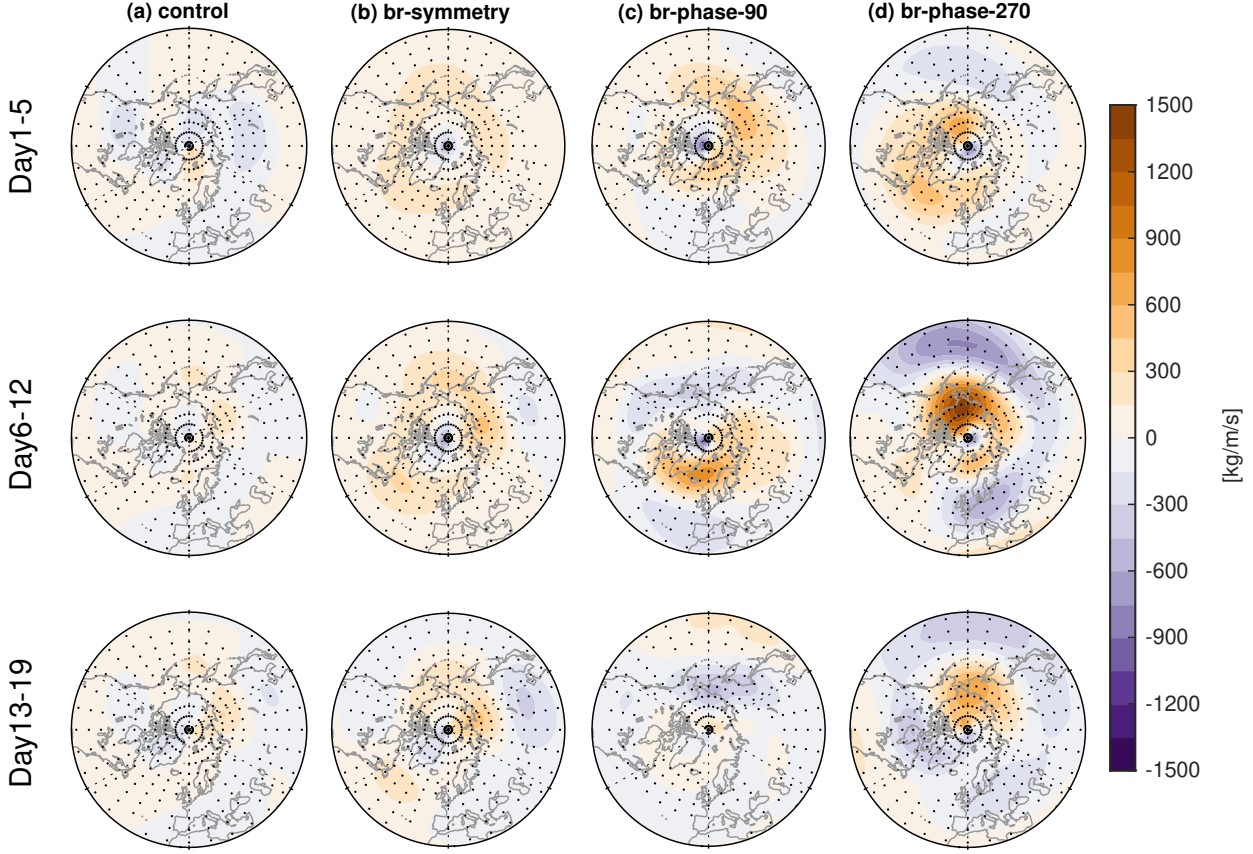

**Figure 7.** Mass streamfunction (derived from divergent part of meridional wind) anomalies pressure-weighted over 34-10 hPa during days 1 to 19 for (a) control run, (b) symmetric branch ensemble, (c) phase-90 ensemble, and (d) phase-270 ensemble. Stippling indicates statistically significant anomalies at the 95% level based on Student's $t$ test. Based on the conventions in section 3, positive anomalies indicate poleward meridional flow, and negative anomalies indicate equatorward flow.

tion in the stratosphere and in the troposphere indicates that the mechanism of White et al. (2022) for zonally symmetric SSWs

accounts for the zonally asymmetric torques also applied here: the troposphere dynamically adjusts to the stratospheric torque, with the biggest impact in the same sector as the applied stratospheric torque.

    Next, to investigate the role of downward coupling in shaping regional surface temperature anomalies, we show the mass streamfunction pressure-integrated over 970-850 hPa in the lower troposphere in Figure 9. Across all experiments, significant equatorward motion is broadly observed around the pole, while poleward motion primarily occurs over subtropical Eurasia.

These circulation patterns peak after the forcing switches off (in contrast to the stratospheric poleward motion, which peaks during the forcing period – Figure 7), which is the same as the tendency of surface temperature anomalies (Figure 4). In particular, equatorward motion facilitates the southward transport of colder air masses from the pole to the subtropics, thus inducing pronounced cooling over Alaska and eastern Eurasia in the phase-90 ensemble (Figure 4 c) and over northern and



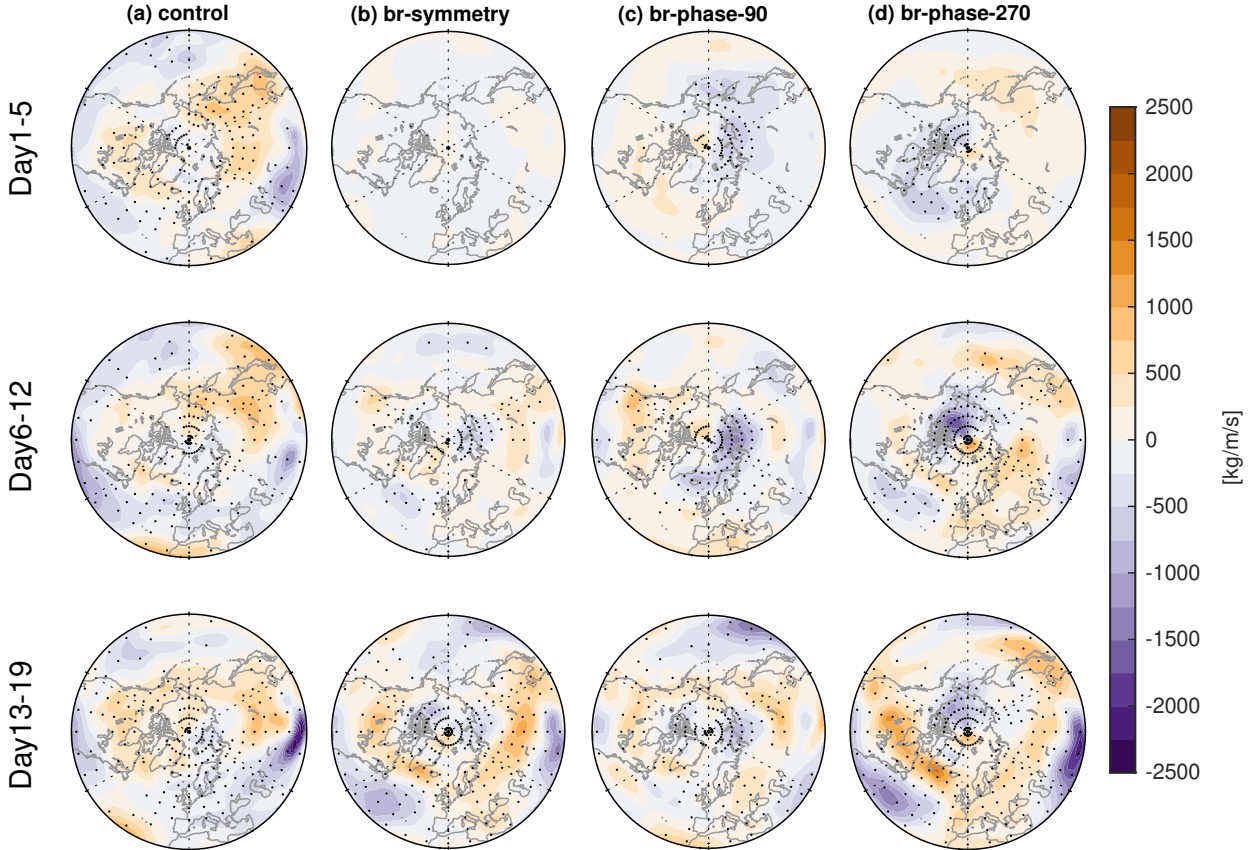

**Figure 8.** As in Figure 7 but over 700-321 hPa.

western Eurasia in the phase-270 ensemble (Figure 4 d). Conversely, poleward motion advects warmer air masses from lower

latitudes toward higher latitudes, but is much more pronounced over Eurasia in the phase-270 ensemble than in the phase-90 ensemble, consistent with the broader and more robust warming over mid-latitude Eurasia in the phase-270 ensemble than the phase-90 ensemble (Figure 4 c,d). Furthermore, the spatially limited poleward flow over southern Europe in the phase-90 ensemble is consistent with the warming signal over Europe evident only in that ensemble (Figure 4 c). Quantifying the importance of this mechanism as compared to the advection of temperature by the anomalous rotational component of wind is

left for future work.

This dynamical interaction between equatorward and poleward motions underscores the important role of the lower-tropospheric meridional circulation in modulating regional surface temperature patterns. While this mechanism is not the sole driver accounting for cooling and warming patterns following SSWs and wave reflection events, it represents a critical pathway through which stratospheric perturbations propagate downward and ultimately influence surface conditions. Overall, this study highlights the

crucial role of the mass streamfunction by the divergent component of meridional wind in facilitating stratosphere-troposphere coupling and, ultimately, significantly leads to surface cooling and warming patterns.





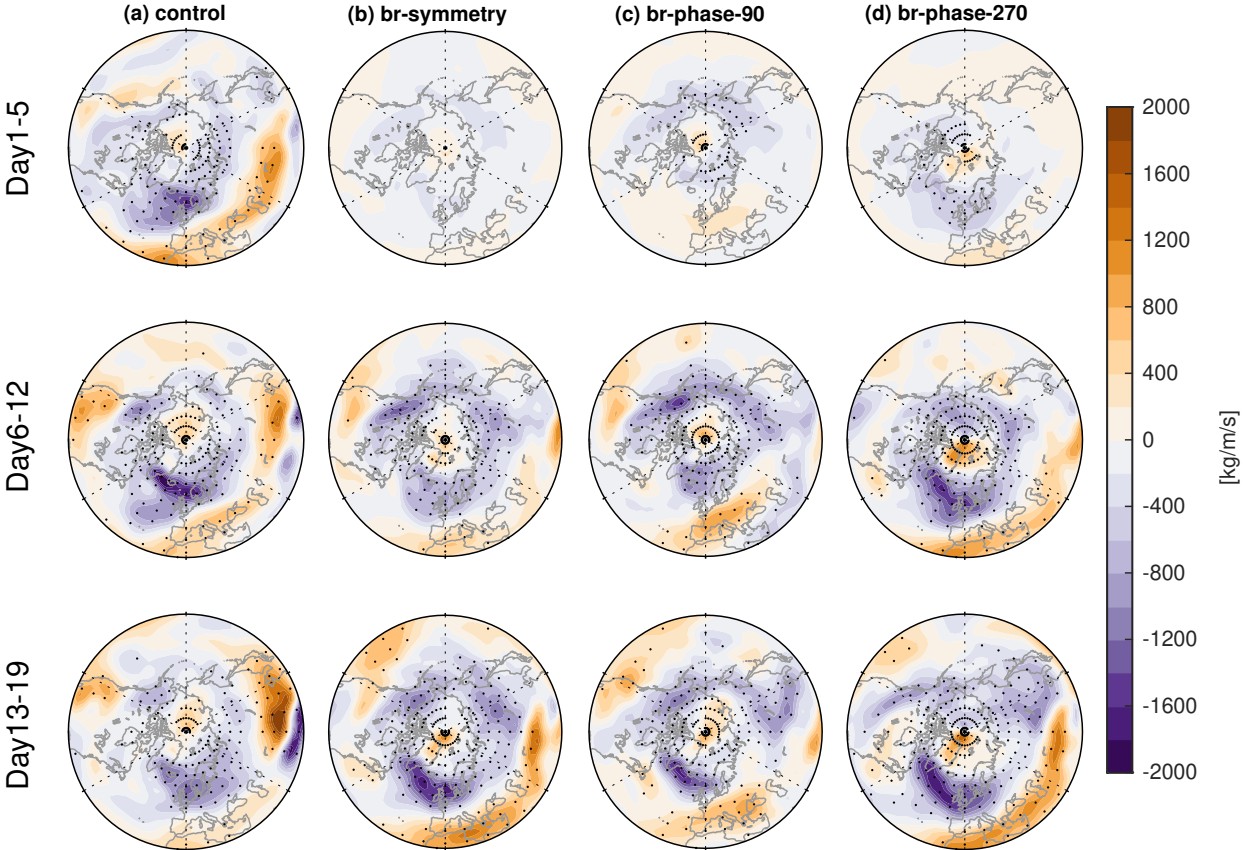

**Figure 9.** As in Figure 7 but over 970-850 hPa.

## 5 Discussion and Conclusions

A better understanding of the surface response to zonal asymmetries of the polar vortex, and specifically of wave reflection during weak vortex events, is important. While previous work using observational data has found evidence that the zonal structure of the vortex anomaly and downward wave propagation matters for the surface impact, it is difficult to isolate the causality using observational data as typically the anomalous vortex state was preceded by a burst of wave activity in the troposphere, and it is not clear how to separate the long-term effects of this initial wave activity from the subsequent stratospheric and tropospheric response.

For example, while both displacement- and split-SSWs affect surface temperatures with potential differences in the spatial pattern and magnitude (Hall et al., 2021; Mitchell et al., 2013), these different SSW morphologies are preceded by qualitatively different initial tropospheric wave fluxes: displacement-SSWs are primarily driven by wave-1 forcing, whereas split-SSWs are influenced by a combination of wave-1 and wave-2 forcings (Bancala et al., 2012; Cohen and Jones, 2011; White et al., 2019). In addition to SSW events, wave reflection events can also significantly impact the surface weather, often leading to extreme





cooling over North America (Cohen et al., 2021, 2022; Matthias and Kretschmer, 2020). But wave reflection events are also
typically preceded by a burst of wave activity over Siberia, and it is not trivial to isolate the importance of the stratospheric re-
flection from the tropospheric wave activity over Siberia for the subsequent North American response. While there are advanced
techniques based on causal effect networks or Granger/Pearl causality that have been applied (Kretschmer et al., 2018a, b),
these techniques ultimately do not allow for a clear mechanistic understanding. Further, the dynamical mechanism(s) account-
ing for the varying surface responses to different types of SSWs and associated wave reflection events remain insufficiently
understood.

To investigate the role of zonal asymmetries of the polar vortex and the wave reflection phenomenon during SSW events,
we extend the approach of White et al. (2022) by adding momentum torques with a strong zonally asymmetric component in
the stratosphere of the MiMA model (Garfinkel et al., 2020a; Jucker and Gerber, 2017). Taking every midnight on 1st January
from the CTRL run as an initial condition, we branch off and impose a momentum torque with various longitudinal phases.
In the present study, we focus on two representative perturbation experiments, phase-90 ensemble ($k = 1$, $\Theta_A = 4$, $\lambda_0 = 90°$)
and phase-270 ensemble ($k = 1$, $\Theta_A = 4$, $\lambda_0 = 270°$). More experiments will be analyzed in a subsequent paper. These branch
ensembles are compared to 48 spontaneously occurring SSWs in CTRL.

The zonally opposite pattern of wave-1 forcing leads to contrasting influences in the stratosphere and troposphere. During
the forcing stage, the zonal mean zonal wind at $60°$N and 10 hPa suddenly transitions from westerlies to easterlies under the
impact of the easterly torque (Figure 3a). Concurrently, the polar cap becomes warmer and exhibits a higher geopotential height
(Figure 3b,c,d), as the enhanced downwelling in the BDC leads to more adiabatic heating (Baldwin et al., 2021; Garfinkel et al.,
2010). In the phase-90 and phase-270 experiments, stratospheric zonal wind anomalies peak in opposite hemispheres, and
this difference in the stratosphere is mirrored by a qualitatively different impact in the troposphere (Figure 1). Furthermore,
the polar vortex shifts away from the pole, but in opposite directions at 10 hPa during days 1 to 5 (Figure 2 c,d) due to
the opposite phasing of the wave-1 component of the forcing. The tropospheric geopotential height anomalies also exhibit
significant displacements from the pole, with the direction of the shift mirroring that in the stratosphere and opposite between
the different experiments. Overall, the opposite phase of wave-1 forcing induces markedly different impacts on the zonal wind,
temperature, and geopotential height fields throughout the stratosphere and troposphere by downward dynamical coupling.

Furthermore, this downward coupling also influences the 2-m temperature and precipitation patterns. Signals of cooling,
warming, and enhanced rainfall appear while the forcing is still on, though their spatial distributions differ among these exper-
iments. The cooling and warming signals intensify well after the stratospheric forcing is turned off (Figure 4), while rainfall
anomalies peak immediately after the forcing is turned off (in days 13 to 19) before subsequently weakening (Figure 5). In
the phase-90 ensemble, cooling is most pronounced over Alaska and eastern Eurasia (Figure 4c), with enhanced rainfall con-
centrated over the North Pacific and the North Atlantic, extending to northwest Europe. (Figure 5c). However, the phase-270
ensemble exhibits significant cooling over central North America and North Eurasia (Figure 4d), with intensified rainfall over
the North Pacific and the North Atlantic, stretching into subtropical Eurasia (Figure 5d). These surface temperature anoma-
lies in phase-90 and phase-270 ensembles are consistent with the observed surface temperature anomalies in clusters 4 and
5 (Kretschmer et al., 2018a). In addition, downward wave propagation occurs over North America in the phase-90 ensemble





(Figure 6c), consistent with that in cluster 4 of Kretschmer et al. (2018a). However, it is concentrated over northern Eurasia
in the phase-270 ensemble (Figure 6d), consistent with the wave reflection events in cluster 5 observed by Kretschmer et al.
(2018a). These results indicate that the opposite phase of wave-1 forcing modulates the SSWs and downward wave propagation
differently, leading to distinct impacts on the surface temperature and precipitation patterns.

We elucidate the pathway of stratosphere-troposphere coupling associated with SSWs and wave reflection events by ana-
lyzing the mass streamfunction of the divergent component of the meridional wind. In all experiments, significant poleward
motion occurs in the stratosphere (Figure 7), while pronounced equatorward motion emerges in the troposphere (Figure 8).
However, the meridional mass streamfunction exhibits opposite zonal dipole patterns between the phase-90 and phase-270
experiments. In the stratosphere while the torque is being applied, poleward motion is predominantly located over the East-
ern Hemisphere in the phase-90 ensemble (Figure 7c), whereas it is centered over the Western Hemisphere in the phase-270
ensemble (Figure 7d). This zonal asymmetry in the stratospheric meridional circulation is mirrored in the free troposphere,
indicating a robust downward coupling between the stratosphere and troposphere. In the lower troposphere (Figure 9), equator-
ward flow transports cooler air from higher to lower latitudes, contributing to surface cooling over Alaska and eastern Eurasia
in the phase-90 ensemble (Figure 4c), and over north Eurasia in the phase-270 ensemble (Figure 4d). Conversely, poleward
flow conveys warmer air from subtropical regions toward the poles, resulting in surface warming over Europe in the phase-90
ensemble (Figure 4c) and over subtropical Eurasia in the phase-270 ensemble (Figure 4d).

Overall, the results of this study bolster the conclusions of previous reanalysis-based studies, which link surface cooling and
warming patterns to stratospheric wave reflection (Cohen et al., 2021, 2022; Kretschmer et al., 2018a). Namely, for the first
time to the best of our knowledge, we have demonstrated that an imposed stratospheric forcing which generates downward
propagating wave activity can causally lead to the surface impacts shown in previous observational work. Our work also
indicates that the direction in which the polar vortex is displaced during an SSW displacement event is of crucial importance
for the surface response. Ongoing work will explore other longitudinal phases for a wave-1 forcing and also wave-2 forcings,
and also better quantify the importance of the meridional divergent flow versus the rotational flow for the surface temperature
impacts.

*Data availability.* The updated version of MiMA used in this study including the modified source code and example name lists to re-
produce the experiments can be downloaded from https://github.com/ianpwhite/MiMA/releases/tag/MiMA-ThermalForcing-v1.0beta (with
DOI: https://doi.org/10.5281/zenodo.4523199). It is expected that these modifications will also eventually be merged into the main MiMA
repository which can be downloaded from https://github.com/mjucker/MiMA.

*Author contributions.* WN and CIG designed the study. WN performed the analysis, produced the figures, and drafted the paper. All authors
discussed the results and edited the paper.



*Competing interests.* The authors declare no conflict of interest.

440 *Acknowledgements.* We thank Eli Galanti and Yohai Kaspi for providing the code used to compute the three-dimensional streamfunction. Wuhan Ning, Chaim I. Garfinkel, and Jian Rao are supported by the ISF–NSFC joint research program (Israel Science Foundation grant no. 3065/23 and National Natural Science Foundation of China grant no. 42361144843). Wuhan Ning, Chaim I. Garfinkel, and Judah Cohen are supported by the NSF–BSF joint research program (National Science Foundation grant no. AGS-2140909 and United States–Israel Binational Science Foundation grant no. 2021714).



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
