# Peer review of "The tropospheric response to zonally asymmetric momentum torques: implications for the downward response to wave reflection and SSW events"

_EGUsphere, 2025_

## Referee Comment (RC1)

**Review of "The tropospheric response to zonally asymmetric momentum torques: implications for the downward response to wave reflection and SSW events" by Ning et al.**

This study uses an intermediate-complexity moist GCM to analyze the impacts of wave-1 momentum torques with varying longitudinal phases. The goal is to help understand the surface influences and mechanisms of wave reflection events. Forced with wave-1 forcings of opposite phases, simulations show distinct circulation and surface responses that are generally consistent with results from the reanalysis in existing literature. The surface air temperature anomalies are further connected to the stratospheric anomalies via mass streamfunction analysis.

The result is within the scope of the journal. The technical aspects of the study are solid, and the paper is generally well written. I believe this study could be accepted for publication after addressing the following issues.

**General comments**

My main concern is that the authors might conflate potentially related yet distinct concepts. The manuscript claims that the results indicate the surface response to stratospheric wave reflection events is causally forced by stratospheric perturbations (L17). Planetary wave reflection is a generally well-defined phenomenon in which planetary waves propagate upward from the troposphere into the stratosphere and are reflected back. Intuitively, upward wave propagation, a reflective surface, and downward wave propagation would be expected in a wave reflection event. The present experiment only shows the downward wave propagation, yet it seems to implicitly imply that downward wave propagation equals wave reflection. I think a discussion is warranted to discuss the model's limitations and clarify the relationship between the results and wave reflection. The authors may also consider softening some related arguments regarding the implications of the results.

The authors claim to be the first to demonstrate that downward wave propagation can causally lead to surface impacts (L86, 427). I am concerned that this might be overstated; at least some of the aspects have been shown in the existing literature. Dunn-Sigouin and Shaw (2018) investigate the mechanism of extreme stratospheric wave-1 negative heat flux events with nudging experiments in a dry dynamical core model. They demonstrate the causal role of stratospheric waves in reproducing the tropospheric responses to negative eddy heat flux events. Through stratospheric nudging experiments in an AGCM, Ding et al. (2023) provide causal evidence that strong stratospheric wave activity leads to North American cold anomalies through vertical wave coupling. Their strong stratospheric wave events exhibit regional raw negative Plumb flux over North America (Fig. 2). They also analyzed weak wave events, during which the stratospheric planetary waves are in an opposite phase compared to those of strong wave events.

Neither of these studies presents exactly the same analysis/results as this study, so I think this analysis is still novel. But the authors should include these publications in their introduction/discussion. And perhaps the point can't be quite so strongly stated.

**Individual comments**

L5: The experiment names "90E" and "270E" do not feel very informative, as they represent the longitude where the zonally asymmetric component is 0. I'm unsure if it is a good idea, but the authors may consider using the longitude of the largest easterly forcing.

L56: Huang et al. (2018) show that the shifting direction of displacement events affects the surface responses in reanalysis.

L58: An imposed heating induces an effect on meridional circulation opposite to what?

L66: Ding et al. (2022) show the distinction between zonally symmetric and asymmetric stratospheric variabilities.

L88: Could you point me to the reference that suggests wave reflection events often occur after SSW events? A single case of 2021 does not read like enough evidence.

L139: easterly -> westerly

L141: The SSW frequency seems much lower than observed: 48 events in 9 runs of 50 years. How so?

L171: Why set the scaling factor of the asmmyeric component to 4? Do other settings change the conclusion?

Figure 2: Why show the geopotential height patterns during days 1 to 5 while showing zonal wind anomalies during days 6 to 12? Do the two periods differ a lot?

Figure 2: I suggest adding contours of 10 hPa anomalies, which would help readers understand through a different lens. Comparing the 10 hPa anomaly with the 500 hPa anomaly would provide insight into the vertical structure.

L245: This is confusing without showing the stratospheric anomalies.

L256: The downward propagation seems hard to see, as the peak anomalies at 500 hPa occur at the same time as those at 10 hPa (day 12) for the branch experiments. In contrast, a clear lag exists between stratospheric and tropospheric anomaly peaks for SSWs in the control experiment.

L265: There is no visible signal during days 1 to 12. How should we see the differences?

L283: Precipitation anomalies following symmetric forcing and control SSWs appear to be more persistent.

- L311, 410: Cluster 5 in the references appears to represent weak polar vortex events. Please elaborate on how they are connected to the downard wave propagation reported here.
- Figure 7: Why do the centers of poleward anomalies seem to shift 90 degrees westward from day 1-5 to day 6-12 in panels (c) and (d)?
- L363: What do you mean by "wave reflection during weak vortex events"? Perlwitz and Harnik (2003) show that a reflective surface is more likely to form when the lower stratospheric polar vortex is anomalously strong. Ding et al. (2022) also present a stronger-than-normal polar vortex prior to strong stratospheric wave events that are associated with regional wave reflection.

**References**

- Ding, X., Chen, G., Sun, L., & Zhang, P. (2022). Distinct North American Cooling Signatures

  Following the Zonally Symmetric and Asymmetric Modes of Winter Stratospheric

  Variability. Geophysical Research Letters, 49(6), e2021GL096076.

  https://doi.org/10.1029/2021GL096076
- Ding, X., Chen, G., Zhang, P., Domeisen, D. I. V., & Orbe, C. (2023). Extreme stratospheric wave activity as harbingers of cold events over North America. Communications Earth & Environment, 4(1), 187. https://doi.org/10.1038/s43247-023-00845-y
- Dunn-Sigouin, E., & Shaw, T. A. (2018). Dynamics of Extreme Stratospheric Negative Heat Flux Events in an Idealized Model. Journal of the Atmospheric Sciences, 75(10), 3521–3540. https://doi.org/10.1175/JAS-D-17-0263.1
- Huang, J., Tian, W., Gray, L. J., Zhang, J., Li, Y., Luo, J., & Tian, H. (2018). Preconditioning of Arctic Stratospheric Polar Vortex Shift Events. Journal of Climate, 31(14), 5417–5436. https://doi.org/10.1175/JCLI-D-17-0695.1
- Perlwitz, J., & Harnik, N. (2003). Observational Evidence of a Stratospheric Influence on the Troposphere by Planetary Wave Reflection. Journal of Climate, 16(18), 3011–3026. https://doi.org/10.1175/1520-0442(2003)016%253C3011:OEOASI%253E2.0.CO;2

---

## Referee Comment (RC2)

Review of 'The tropospheric response to zonally asymmetric momentum torques: implications for the downward response to wave reflection and SSW events' by Ning et al.

**General comments:**

This study uses an intermediate-complexity GCM (MiMA) with imposed stratospheric momentum torques to investigate how zonally asymmetric forcing affects the tropospheric and surface response following SSWs and wave reflection events. By comparing symmetric forcing with different wave-1 forcings, the authors isolate the causal impact of stratospheric zonal structure. The manuscript is generally well written and the analyses are systematic. However, the current manuscript lacks integration across sections, which weakens the logical flow. In addition, a clearer illustration of the experimental design is needed. Therefore, I recommend a major revision.

**Major Comments:**

- 1. Improve integration and narrative flow. While individual sections provide informative analyses, they are currently disconnected. This is reflected in several aspects:
  - First, the introduction section heavily focuses on the SSWs, while wave reflection receives less attention, despite both being central in the title and abstract. In my opinion, a more balanced structure is needed, perhaps starting with general features and influence of stratospheric polar vortex variability would better frame the study.
  - Second, the analyses of SSW-like events and wave reflections appear as separate topics, although they arise from the same experiments. This suggests that these SSWs and wave reflections are dynamically linked. One implication could be that SSWs are associated with different types of wave reflections depending on the phase of the imposed stratospheric wave forcing, which itself is an important conclusion worth highlighting.
  - Third, the surface response results from both the wave reflection and the downward coupling discussed in Section 4.4. However, in the current format, these aspects are presented rather separately.

I highly suggest adjusting the structure to make the story more coherent. For instance, emphasize that multiple mechanisms can together explain the surface response (wave reflection, mass streamfunction). Alternatively, the authors could first introduce and discuss the features and dynamical processes, and then link them to the surface response.

2. Clarify and illustrate the experimental design. The description of the model experiments is a bit hard to follow. There are several groups of experiments, the control runs (9 runs times 50 years), the CTRL run (the median gravity wave drag one, 50 years), and the branch

experiments (50 years of the CTRL run times different forcings). This information is described across multiple paragraphs and I had to re-read Section 2 several times to fully understand it. To help readers quickly grasp the essentials of the model settings, I suggest adding a schematic diagram or a concise summary table to visually summarize the experimental setup.

**Specific Comments:**

- 1. Introduction: The Introduction currently spends many paragraphs on SSWs while only briefly mentioning wave reflections. Since both are emphasized in title and abstract, I suggest the authors to restructure it to (i) introduce the general polar vortex variability and its tropospheric impact, (ii) introduce SSWs and wave reflection in turn and their linkage, and (iii) identify the gaps that this study aims to address. This will provide a more balanced background and better motivate the focus of this work.
- 2. L50: 'cleanly' should be 'clearly'.
- 3. L142: by 'In all, 48 SSW events are identified across all nine control runs during the JFM period', do you mean 48 SSWs in total over the 9 runs\*50 years (i.e., 450 winters)? Please clarify. If so, this implies a very low frequency of SSWs even only JFM is considered, which may warrant brief discussion.
- 4. Figure 1: the color bar in the upper panel spans from -120 to 120m/s. Are the zonal wind anomalies really this large in your experiments? Please clarify whether this reflects the imposed forcing amplitude and discuss whether the results are sensitive to the forcing amplitude.
- 5. Figure 1 and Figure 2 focus on different period, why is that? Also, should the unit in Figure 2 should be gpm?
- 6. Section 4.2. Figure 4 suggests that the peak surface response amplitude is similar across experiments, but the control run response occurs earlier. Please discuss the possible reason. I also suggest the authors including a time-height evolution of the NAM index or zonal wind anomaly to illustrate the zonal-mean downward propagation, which might be helpful to understand the different surface response.

- 7. Figure 4 and 5. I suggest the authors including the tropospheric circulation to briefly compare if the circulation differs among different experiments. Even a brief illustration would help clarify the connection between stratospheric forcing and surface patterns.
- 8. L303-304: In the symmetry ensemble, the Fz is not centrally located above the pole, but shows downward propagation over NA in Day 1-12. This might be related to the climatological wave propagation.

---

## Author Comment (AC1)

**Response to the comments from Referee-1**

We sincerely appreciate your thoughtful questions and valuable suggestions, especially the comments regarding wave reflection events during weak polar vortex states or SSWs. We have carefully addressed each of your comments point by point (highlighted in blue) and made the corresponding revisions in the revised version of the manuscript.

Note that there are five main modifications in the revised manuscript: 1) a new structure of the introduction section; 2) a new Table 1 summarizing the analyzed experiments in this study; 3) a new Figure 2 showing the period in days from 6 to 12; 4) a new Figure 6 showing the longitude-pressure cross section of Plumb flux (Fx, Fz); 5) a new Figure 10 showing the zonal wind anomalies at 321 hPa.

**General comments**

My main concern is that the authors might conflate potentially related yet distinct concepts. The manuscript claims that the results indicate the surface response to stratospheric wave reflection events is causally forced by stratospheric perturbations (L17). Planetary wave reflection is a generally well-defined phenomenon in which planetary waves propagate upward from the troposphere into the stratosphere and are reflected back. Intuitively, upward wave propagation, a reflective surface, and downward wave propagation would be expected in a wave reflection event. The present experiment only shows the downward wave propagation, yet it seems to implicitly imply that downward wave propagation equals wave reflection. I think a discussion is warranted to discuss the model's limitations and clarify the relationship between the results and wave reflection. The authors may also consider softening some related arguments regarding the implications of the results.

We agree that our methodology is not intended to study the entire lifecycle of wave reflection events - indeed the entire point is to isolate the downward propagation part from the antecedent upward pulse, as the antecedent upward pulse necessarily entails strong tropospheric anomalies. In general, we tried to be careful in our wording, but L17 needed to be clarified. Sorry for the confusion. But we want to make clear that there is unambiguous wave reflection in these experiments.

**Figure R1**. Longitude-pressure section of raw Fx and Fz (arrows; zonal and vertical components of Plumb flux) and raw geopotential height eddy (shadings; remove the zonal mean) area averaged between 55°N and 80°N in days 6-12.

Figure R1 shows the longitude-pressure section of raw Fx and Fz and raw geopotential height eddy area-averaged between 55°N and 80°N in days 6-12. The planetary waves propagate upward and are reflected back mainly over North America in the symmetry and phase-90 ensembles (Fig. R1 b,c) but dominantly over North Eurasia in the phase-270 ensemble (Fig. R1 d), which is consistent with the Fz in Figure 7 of the manuscript. In the wave reflection regions, the geopotential height eddy shows an eastward tilt, which also indicates a wave reflection event. We want to highlight the impact of zonal asymmetries of the polar vortex on the zonal asymmetries of wave reflection events. Therefore, we also show the map of Fz in this manuscript. Figure R1 above is now included in Figure 6 in the revised manuscript.

Zonal asymmetries of the wave reflection event are closely associated with the zonal asymmetries of the stratospheric polar vortex. We use the mass streamfunction to reveal the downward dynamical coupling from the stratosphere to the troposphere due to the additive effects from the polar vortex and wave reflection event, because they are dynamically coupled with each other.

We add some discussion on the limitations of MiMA starting from L132.

Note that wave reflection events could happen during both strong and weak polar vortex states, and we focus on the wave reflection event during SSWs in this study. Please see Table A1 in the corrigendum of Messori et al. (2022). It shows 45 selected stratospheric reflection events and clarifies 9 reflection events associated with SSWs from 1980 to 2021.

The authors claim to be the first to demonstrate that downward wave propagation can causally lead to surface impacts (L86, 427). I am concerned that this might be overstated; at least some of the aspects have been shown in the existing literature. Dunn-Sigouin and Shaw (2018) investigate the mechanism of extreme stratospheric wave-1 negative heat flux events with nudging experiments in a dry dynamical core model. They demonstrate the causal role of stratospheric waves in reproducing the tropospheric responses to negative eddy heat flux events. Through stratospheric nudging experiments in an AGCM, Ding et al. (2023) provide causal evidence that strong stratospheric wave activity leads to North American cold anomalies through vertical wave coupling. Their strong stratospheric wave events exhibit regional raw negative Plumb flux over North America (Fig. 2). They also analyzed weak wave events, during which the stratospheric planetary waves are in an opposite phase compared to those of strong wave events.

Neither of these studies presents exactly the same analysis/results as this study, so I think this analysis is still novel. But the authors should include these publications in their introduction/discussion. And perhaps the point can't be quite so strongly stated.

Thank you for sharing these valuable papers. We have carefully compared them with our results and cited these papers in the suitable places of the revised manuscript.

**Individual comments**

L5: The experiment names "90E" and "270E" do not feel very informative, as they represent the longitude where the zonally asymmetric component is 0. I'm unsure if it is a good idea, but the authors may consider using the longitude of the largest easterly forcing.

The location of the largest easterly forcing is not consistent with the center of raw Z10 in Figure 2. Therefore, we remove this part "centered at 90E vs. 270E" from the abstract, just using the "phase-90" or "phase-270" to name the two experiments by specifying the longitudinal phase and directly linking them to equation 4.

L56: Huang et al. (2018) show that the shifting direction of displacement events affects the surface responses in reanalysis.

Cited near L32.

L58: An imposed heating induces an effect on meridional circulation opposite to what?

An imposed heating would induce a qualitatively opposite effect on the meridional circulation in the naturally occurring SSWs. In naturally occurring SSWs, the strengthened downwelling induces intensified adiabatic heating and results in stronger poleward meridional circulation in the stratosphere, while the imposed heating would instead produce upwelling and thereby equatorward meridional circulation in the stratosphere.

L66: Ding et al. (2022) show the distinction between zonally symmetric and asymmetric stratospheric variabilities.

**Cited in L33.**

L88: Could you point me to the reference that suggests wave reflection events often occur after SSW events? A single case of 2021 does not read like enough evidence.

Please see Table A1 in the corrigendum of Messori et al. (2022). It shows 45 selected stratospheric reflection events and clarifies 9 reflection events associated with SSWs from 1980 to 2021.

L139: easterly -> westerly

**Corrected.**

L141: The SSW frequency seems much lower than observed: 48 events in 9 runs of 50 years. How so?

We focus on only the JFM period, excluding November and December, which would give a lower ratio. White et al. (2022) reported a ratio of 0.29 SSWs per year in their control runs by using MiMA. This ratio is still smaller than the observed ~0.67 SSWs per year, while slightly different gravity wave settings lead to an SSW frequency above 0.4 per year.

This discussion has been added to L132 in the revised manuscript.

L171: Why set the scaling factor of the asymmetric component to 4? Do other settings change the conclusion?

We set the scaling factor to 4 to allow for a substantial zonally asymmetric component of the vortex response. Lowering it leads to a weaker regional signal, and strengthening it leads to a stronger regional signal.

We have also performed experiments with a wide range of zonally asymmetric scaling factors, though for all but the final configuration the model output wasn't retained, and plots weren't saved for most configurations either. But to give the reviewer a sense that results are insensitive, below we include a figure (Figure R2) showing the Z 500hpa anomalies in week 1 for

a scaling factor of 0.8 (the paper uses 4; Figure R3) for the zonally asymmetric component. The anomalies are very linear. A larger scaling factor amplifies regional anomalies, while the spatial pattern of responses is essentially identical in Figures R2 and R3. Note that the anomalies in Figures R2 and R3 are the differences between phase-90/270 and symmetry ensembles.

This discussion has been added to L181 in the revised manuscript.

**Figure R2.** Geopotential height anomalies (m) at 500 hPa in week 1 with scaling factor 0.8. Note that the anomalies are the differences between phase-90/270 and symmetry ensembles.

**Figure R3.** Geopotential height anomalies (m) at 500 hPa in week 1 with scaling factor 4. Note that the anomalies are the differences between phase-90/270 and symmetry ensembles.

Figure 2: Why show the geopotential height patterns during days 1 to 5 while showing zonal wind anomalies during days 6 to 12? Do the two periods differ a lot?

There are no big differences between days 1-5 and days 6-12, except for the more intensified anomalies in days 6-12. We have replaced the days 1-5 panels in Figure 2 with the geopotential height patterns during days 6 to 12 in the revised manuscript.

Figure 2: I suggest adding contours of 10 hPa anomalies, which would help readers understand through a different lens. Comparing the 10 hPa anomaly with the 500 hPa anomaly would provide insight into the vertical structure.

L245: This is confusing without showing the stratospheric anomalies.

Good suggestion, please see the new Figure 2 in the revised manuscript.

L256: The downward propagation seems hard to see, as the peak anomalies at 500 hPa occur at the same time as those at 10 hPa (day 12) for the branch experiments. In contrast, a clear lag

exists between stratospheric and tropospheric anomaly peaks for SSWs in the control experiment.

The external forcings are applied only above 100 hPa in the stratosphere. Consequently, the tropospheric anomalies arise from the downward coupling of the stratospheric perturbations. On shorter timescales (e.g., 6-hourly), the tropospheric response lags behind the stratospheric anomalies.

L265: There is no visible signal during days 1 to 12. How should we see the differences?

The cooling or warming signal is weak and not significant in days 1-12, but the spatial pattern is consistent with that occurring in days 13-33 for each branch ensemble, and the spatial pattern differs a lot between different ensembles, especially between phase-90 and phase-270 ensembles.

L283: Precipitation anomalies following symmetric forcing and control SSWs appear to be more persistent.

This makes sense. The composite of control SSWs shows a similar geopotential height pattern as the symmetry ensemble (as seen from Figure 2) because it includes many different types of SSWs with different zonal asymmetries. Here, we modify this sentence to be "precipitation anomalies weaken rapidly in phase-90 and phase-270 ensembles" in the revised manuscript.

L311, 410: Cluster 5 in the references appears to represent weak polar vortex events. Please elaborate on how they are connected to the downward wave propagation reported here.

Figure 2d of Kretschmer et al. (2018) shows the anomalous Fz at 100 hPa for cluster 5. Anomalous negative Fz emerges over both the east and west of North Eurasia, which indicates the wave reflection events there. Cohen et al. (2021) only presented the Fz at 100 hPa for cluster 4 in their Figure 3, although we discussed with Cohen the results of cluster 5 offline. This paper (Cohen et al. 2021) has been removed from here in the revised manuscript.

Figure 7: Why do the centers of poleward anomalies seem to shift 90 degrees westward from day 1-5 to day 6-12 in panels (c) and (d)?

**Figure R4.** Anomalies of the divergent part of the meridional wind at 10 hPa for (a) control run, (b) symmetric branch ensemble, (c) phase-90 ensemble, and (d) phase-270 ensemble. Stippling indicates statistically significant anomalies at the 95% level based on Student's t test.

Figure R4 shows the anomalies of the divergent part of meridional winds at 10 hPa. In days 6-12, the positive anomalies of the divergent part of meridional wind are intensified in both the phase-90 and phase-270 ensembles (Figure R4 c,d; middle row), while the negative anomalies emerge in the sector opposite to the positive anomalies, resulting in anomalous downwelling there, thereby intensifying the anomalous high at 10 hPa (Figure 2 in the manuscript). Therefore, the shift of the center of poleward anomalies is due to the development of the polar vortex.

L363: What do you mean by "wave reflection during weak vortex events"? Perlwitz and Harnik (2003) show that a reflective surface is more likely to form when the lower stratospheric polar vortex is anomalously strong. Ding et al. (2022) also present a stronger-than-normal polar vortex prior to strong stratospheric wave events that are associated with regional wave reflection.

The wave reflection event could happen during both strong and weak polar vortex states, and we focus on the wave reflection event during SSWs in this study. Please see Table A1 in the corrigendum of Messori et al. (2022). It shows 45 selected stratospheric reflection events and clarifies 9 reflection events associated with SSWs from 1980 to 2021.

**References**

- Judah Cohen et al. ,Linking Arctic variability and change with extreme winter weather in the United States.Science373,1116-1121(2021).DOI:10.1126/science.abi9167
- Kretschmer, M., Cohen, J., Matthias, V., Runge, J., and Coumou, D.(2018): The different stratospheric influence on cold-extremes in Eurasia and North America, npj Climate and Atmospheric Science, 1, 44, 2018. <a href="https://doi.org/10.1038/s41612-018-0054-4">https://doi.org/10.1038/s41612-018-0054-4</a>
- Messori, G., Kretschmer, M., Lee, S. H., and Wendt, V. (2022): Stratospheric downward wave reflection events modulate North American weather regimes and cold spells, Weather Clim. Dynam., 3, 1215–1236, 2022. https://doi.org/10.5194/wcd-3-1215-2022
- White, I. P., C. I. Garfinkel, and P. Hitchcock, (2022): On the Tropospheric Response to Transient Stratospheric Momentum Torques. J. Atmos. Sci., 79, 2041–2058, <a href="https://doi.org/10.1175/JAS-D-21-0237.1">https://doi.org/10.1175/JAS-D-21-0237.1</a>

---

## Author Comment (AC2)

**Response to the comments from Referee-2**

We sincerely appreciate your thoughtful questions and valuable suggestions, especially the suggestions regarding the structure of the introduction and highlighting the dynamic linkage between SSWs and wave reflections. We have carefully addressed each of your comments point by point (highlighted in blue) and made the corresponding revisions in the revised version of the manuscript.

Note that there are five main modifications in the revised manuscript: 1) a new structure of the introduction section; 2) a new Table 1 summarizing the analyzed experiments in this study; 3) a new Figure 2 showing the period in days from 6 to 12; 4) a new Figure 6 showing the longitude-pressure cross section of Plumb flux (Fx, Fz); 5) a new Figure 10 showing the zonal wind anomalies at 321 hPa.

**General comments:**

This study uses an intermediate-complexity GCM (MiMA) with imposed stratospheric momentum torques to investigate how zonally asymmetric forcing affects the tropospheric and surface response following SSWs and wave reflection events. By comparing symmetric forcing with different wave-1 forcings, the authors isolate the causal impact of stratospheric zonal structure. The manuscript is generally well written and the analyses are systematic. However, the current manuscript lacks integration across sections, which weakens the logical flow. In addition, a clearer illustration of the experimental design is needed. Therefore, I recommend a major revision.

**Major Comments:**

- 1. Improve integration and narrative flow. While individual sections provide informative analyses, they are currently disconnected. This is reflected in several aspects:
- First, the introduction section heavily focuses on the SSWs, while wave reflection receives less attention, despite both being central in the title and abstract. In my opinion, a more balanced structure is needed, perhaps starting with general features and influence of stratospheric polar vortex variability would better frame the study.
- Second, the analyses of SSW-like events and wave reflections appear as separate topics, although they arise from the same experiments. This suggests that these SSWs and wave reflections are dynamically linked. One implication could be that SSWs are associated with different types of wave reflections depending on the phase of the imposed stratospheric wave forcing, which itself is an important conclusion worth highlighting.

• Third, the surface response results from both the wave reflection and the downward coupling discussed in Section 4.4. However, in the current format, these aspects are presented rather separately.

I highly suggest adjusting the structure to make the story more coherent. For instance, emphasize that multiple mechanisms can together explain the surface response (wave reflection, mass streamfunction). Alternatively, the authors could first introduce and discuss the features and dynamical processes, and then link them to the surface response.

Great suggestions! We focus more attention on wave reflection events in the introduction and highlight the additive impacts on surface responses from both SSWs and concurrent wave reflection events, as explained by mass streamfunction and their dynamical linkage, driven by the imposed forcing in the revised manuscript.

- 1. Two paragraphs introduce the wave reflection event starting from L53 in the revised manuscript.
- 2. Highlighting the dynamical linkage between SSWs and wave reflection due to the phasing of imposed forcing in L1-L2, L330-L331, L445-L449, and L467-L472.
- 3. Highlighting the joint impacts on surface response from the SSW and the wave reflection event in L19-L20, L393-L395, and L467-L472.
- 2. Clarify and illustrate the experimental design. The description of the model experiments is a bit hard to follow. There are several groups of experiments, the control runs (9 runs times 50 years), the CTRL run (the median gravity wave drag one, 50 years), and the branch experiments (50 years of the CTRL run times different forcings). This information is described across multiple paragraphs and I had to re-read Section 2 several times to fully understand it. To help readers quickly grasp the essentials of the model settings, I suggest adding a schematic diagram or a concise summary table to visually summarize the experimental setup.

Good suggestion! Please see Table 1 here and also in the revised manuscript.

**Table 1.** List of experiments analyzed in this study. Note that the symmetry, phase-90, and phase-270 ensembles are branched off from the CTRL.

| Name      | Ensemble members | Total period         | Forcing                                                                                           |  |
|-----------|------------------|----------------------|---------------------------------------------------------------------------------------------------|--|
| control   | 9 (8+CTRL)       | $9 \times 50$ years  | free running (differing in gravity wave drag in yr-1; CTRL with median drag)                      |  |
| symmetry  | 50               | $50 \times 4$ months | $M_S = -15 \text{ m s}^{-1} \text{ day}^{-1}$                                                     |  |
| phase-90  | 50               | $50 \times 4$ months | $k = 1, \Theta_{\rm A} = 4, \lambda_0 = 90^{\circ}, M_S = -15 \text{ m s}^{-1} \text{ day}^{-1}$  |  |
| phase-270 | 50               | $50 \times 4$ months | $k = 1, \Theta_{\rm A} = 4, \lambda_0 = 270^{\circ}, M_S = -15 \text{ m s}^{-1} \text{ day}^{-1}$ |  |

**Specific Comments:**

1. Introduction: The Introduction currently spends many paragraphs on SSWs while only briefly mentioning wave reflections. Since both are emphasized in title and abstract, I suggest the

authors to restructure it to (i) introduce the general polar vortex variability and its tropospheric impact, (ii) introduce SSWs and wave reflection in turn and their linkage, and (iii) identify the gaps that this study aims to address. This will provide a more balanced background and better motivate the focus of this work.

Thank you for the valuable suggestions! We put more attention on the wave reflection events and highlight the additive impacts from both SSWs and concurrent wave reflection events in the introduction and conclusion. Please see them in the revised manuscript.

2. L50: 'cleanly' should be 'clearly'.

**Corrected.**

3. L142: by 'In all, 48 SSW events are identified across all nine control runs during the JFM period', do you mean 48 SSWs in total over the 9 runs\*50 years (i.e., 450 winters)? Please clarify. If so, this implies a very low frequency of SSWs even only JFM is considered, which may warrant brief discussion.

Yes, we obtained 48 SSWs across 450 winters. White et al. (2022) reported a ratio of 0.29 SSWs per year in their control runs by using MiMA. This ratio is still smaller than the observed  $^{\circ}$ 0.67 SSWs per year, while slightly different gravity wave settings lead to an SSW frequency above 0.4 per year.

This discussion has been added to L132 in the revised manuscript.

4. Figure 1: the color bar in the upper panel spans from -120 to 120m/s. Are the zonal wind anomalies really this large in your experiments? Please clarify whether this reflects the imposed forcing amplitude and discuss whether the results are sensitive to the forcing amplitude.

Yes, the zonal wind anomalies reach these large values in our experiments due to the strong symmetric component of forcing, -15 m/s/day. The raw U1060 values are of similar magnitude to some of the stronger observed events, but well within the range of observations (see Figure 3 in our manuscript). White et al (2022) examined the linearity of the surface impact from the strength of the stratospheric symmetric component of forcing and found strong linearity. Please see Figure 4 in White et al. (2022), who examined the responses of zonal-mean zonal winds at 10 hPa and 60°N and polar cap temperatures to different symmetric forcings, which shows that a larger forcing leads to weaker zonal winds and warmer polar cap. For this paper, we choose a forcing that is on the strong end of observations to get a strong surface response.

We have also performed experiments with a wide range of zonally asymmetric scaling factors, though for all but the final configuration the model output wasn't retained, and plots weren't saved for most configurations either. But to give the reviewer a sense that results are insensitive, below we include a figure (Figure R1) showing the Z 500hpa anomalies in week 1 for a scaling factor of 0.8 (the paper uses 4; Figure R2) for the zonally asymmetric component. The anomalies are very linear. A larger scaling factor amplifies regional anomalies, while the spatial pattern of responses is essentially identical in Figures R1 and R2. Note that the anomalies in Figures R1 and R2 are the differences between phase-90/270 and symmetry ensembles.

This discussion has been added to L181 in the revised manuscript.

**Figure R1.** Geopotential height anomalies (m) at 500 hPa in week 1 with scaling factor 0.8 for phase-90 and phase-270 ensembles. Note that the anomalies here are differences between phase-90/270 and symmetry ensembles.

**Figure R2.** Geopotential height anomalies (m) at 500 hPa in week 1 with scaling factor 4 for phase-90 and phase-270 ensembles. Note that the anomalies here are differences between phase-90/270 and symmetry ensembles.

5. Figure 1 and Figure 2 focus on different period, why is that? Also, should the unit in Figure 2 should be gpm?

There are no big differences between days 1-5 and days 6-12, except for the stronger anomalies in days 6-12. We replace Figure 2 with the geopotential height patterns during days 6-12 in the revised manuscript to keep it consistent with Figure 1.

Yes, it is better to use the gpm as the unit of geopotential height, although 1 gpm is approximately 1 m. However, we can not replot Figure 3 to modify the unit of geopotential height now because the data disk is offline from our cluster. To keep in consistency with other figures, we continue using "m" as the unit of geopotential height in the revised manuscript. We will modify this issue once the data disk is back.

6. Section 4.2. Figure 4 suggests that the peak surface response amplitude is similar across experiments, but the control run response occurs earlier. Please discuss the possible reason. I also suggest the authors including a time-height evolution of the NAM index or zonal wind

anomaly to illustrate the zonal-mean downward propagation, which might be helpful to understand the different surface response.

In control runs, day 0 indicates the reversal of zonal-mean zonal wind at 10 hPa and 60°N (Figure 3a), transitioning from westerly to easterly. Therefore, days 1-5 are exactly the end of the forcing stage of SSWs in control runs, while days 1-5 are the initial forcing stage in branch ensembles. Consequently, the control run response occurs earlier in days 1-5. We also clarify this in L252 and in L281 in the revised manuscript.

The time-height evolution of the NAM index or zonal wind anomaly can provide only the information when the stratospheric anomalies propagate downward into the lower troposphere without showing the information of regional anomalies, while Figures 4 and 5 show that significant surface anomalies emerge in days 13-19. To better understand the surface temperature response, we suggest linking t2m anomalies (Figure 4) with the mass streamfunction anomalies in the lower troposphere (Figure 11).

We add the geopotential height anomalies at 10 hPa in the new Figure 2. Tropospheric anomalies lie directly beneath the strongest anomalies in the mid-stratosphere but show little westward shift. We believe this could help better understand the downward coupling.

7. Figures 4 and 5. I suggest the authors including the tropospheric circulation to briefly compare if the circulation differs among different experiments. Even a brief illustration would help clarify the connection between stratospheric forcing and surface patterns.

We add the zonal wind anomalies at 321 hPa in the new Figure 10. Please check the relevant discussion starting from L357 in the revised manuscript.

8. L303-304: In the symmetry ensemble, the Fz is not centrally located above the pole, but shows downward propagation over NA in Day 1-12. This might be related to the climatological wave propagation.

Yes, Messori et al. (2022) reported a climatological downward wave propagation over Canada in DJFM periods from 1979 to 2021 based on ERA5 reanalysis (see their Figure 1). This discussion has been included in L317 in the revised manuscript.

**References**

Messori, G., Kretschmer, M., Lee, S. H., and Wendt, V. (2022): Stratospheric downward wave reflection events modulate North American weather regimes and cold spells, Weather Clim. Dynam., 3, 1215–1236, 2022. https://doi.org/10.5194/wcd-3-1215-2022

White, I. P., C. I. Garfinkel, and P. Hitchcock, (2022): On the Tropospheric Response to Transient Stratospheric Momentum Torques. J. Atmos. Sci., 79, 2041–2058, https://doi.org/10.1175/JAS-D-21-0237.1

---

## Referee Report (RR1)

**Review of 'The tropospheric response to zonally asymmetric momentum torques: implications for the downward response to wave reflection and SSW events' by Ning et al.**

**General comments:**

I thank the authors for their detailed responses to the comments made by me and the other reviewer. The revised version shows significant improvement and addresses most of my previous concerns. However, a few statements remain imprecise or overly strong, and adjusting them would help ensure that the conclusions accurately reflect what can be supported by the analyses. Please see the detailed comments below.

**Major Comments:**

1. Some of the statements regarding the causality are too strong. The manuscript frequently uses expressions such as 'causally'. I understand that the main aim is to isolate the influence from the stratospheric forcing, and the authors wish to emphasize this aspect. However, it might be misleading in several contexts. For instance, the last sentence in the abstract, while it is true that the tropospheric response in these experiments originates from the stratospheric forcing, the sentence 'the observed surface response … causally forced by, stratospheric perturbations' (L18-20) is not precise. It is also unclear whether 'observed' refers to the detected model response or observational data. If the latter, the statement is too strong. I suggest rephrasing this as 'can be forced by …' or something similar. In addition, the divergent mass streamfunction can help explain the surface temperature response, but only to a certain extent. The phrase 'causally linked' (L17) might overstate the role of this diagnostic. I recommend softening the tone so that the conclusions more precisely reflect what can be inferred from the present analysis.

2. The statement regarding the consistency with observation/reanalysis is not accurate. The manuscript currently states that the two phases correspond to clusters 4 and 5 of Kretschmer et al. (2018a). However, while the surface temperature response indeed show similarities, the stratospheric circulation does not align in the same way. In particular, wave reflection only appears in cluster 4, where cluster 5 shows a reduced upward wave propagation, but the raw Fz is still positive over Eurasia, unlike the negative raw Fz in the phase-270 experiments here. I understand that we cannot expect the idealized experiments to reproduce every observational feature. However, the comparison should be presented more carefully to avoid implying equivalence where the mechanisms differ. I suggest refining the relevant statements accordingly. In addition, repeatedly referring to cluster numbers may confuse readers unfamiliar with the cited work; it may be clearer to describe their defining characteristics when first introduced and avoid relying solely on "cluster 4/5" labels thereafter.

3. The alignment of timing between branch ensembles and control runs. While the authors noted that 'there is no expectation for the timing of the surface responses to match', the magnitude of response in the branch ensembles after day 13 appears more comparable to those in the control runs after day1. I understand that 'day0' represents different reference points, and strict alignment is not required. But align the timelines based on the peak zonal-wind reversal (e.g., day 0 in the control run and day 12 in the branch runs) may make the comparison more straightforward for readers. Alternatively, omitting direct cross-experiment comparisons at fixed lags, or explicitly noting their limitations, would avoid potential confusion.

**Specific Comments:**

1. L17. Should 'downward propagation events' refer instead to 'wave reflection events'?

2. L61-63 and L408-410. Previous studies have shown that this type of stratospheric anomaly is linked to preceding tropospheric circulation (e.g., Shen et al. 2023; Tan and Bao 2020) and that similar stratospheric disturbances can lead to distinct surface response depending on the tropospheric processes involved (e.g., Shen et al. 2025). Adding a brief discussion where relevant can be helpful to strengthen the motivation for isolating the role of stratosphere.

3. L152. Change 'present' to 'represent'.

4. L203. Should be 'McIntyre' and 'Edmon et al.'.

5. L231. It is more accurate to state 'averaged over days 6 to 12'. The same applies to other similar descriptions.

6. Figure 3c and d. The tropospheric polar-cap height anomaly peaks almost simultaneously with the stratospheric anomaly. Could the authors clarify why this occurs?

7. L273-275. For the reasons described in major comment #2, I suggest reducing the emphasis on direct comparison with Kretschmer et al. (2018a), particularly for phase-270, which does not closely resemble cluster 5 beyond the surface temperature pattern.

8. L283-284. Here the comparison uses days 1-5, but earlier the authors note that the timings are not expected to match. As mentioned in major comment #3, aligning the timing or avoiding such direct comparisons may reduce confusion.

9. L310-312. Please specify the longitude range of the region discussed for easier interpretation.

10. L321-326. Phase-90 shares characteristics with cluster 4, but phase-270 does not resemble cluster5. Revising this statement for accuracy would be beneficial.

11. L445. Cluster 5 in Kretschmer 2018a does not show a wave reflection. This should be corrected.

Reference:

Shen, X., Wang, L., Scaife, A. A., & Hardiman, S. C. (2025). Intraseasonal linkages of winter surface air temperature between Eurasia and North America. Geophysical Research Letters, 52, e2024GL113301. https://doi.org/10.1029/2024GL113301

Tan, X., and M. Bao, 2020: Linkage between a dominant mode in the lower stratosphere and the Western Hemisphere circulation pattern. Geophys. Res. Lett., 47, e2020GL090105, https://doi.org/10.1029/2020GL090105

Shen, X., Wang, L., Scaife, A. A., Hardiman, S. C., & Xu, P. (2023). The Stratosphere-Troposphere Oscillation as the dominant intraseasonal coupling mode between the stratosphere and troposphere. Journal of Climate, 36(7), 2259–2276. https://doi.org/10.1175/jcli-d-22-0238.1

---

## Author Response (AR2)

**Response to the comments from Referee-2**

We sincerely appreciate your thoughtful questions and valuable suggestions, especially the suggestions regarding carefully concluding our results and reducing the comparison with cluster 5 of Kretschmer et al. (2018a). We have carefully addressed each of your comments point by point (highlighted in blue) and made the corresponding revisions in the revised version of the manuscript.

**General comments:**

I thank the authors for their detailed responses to the comments made by me and the other reviewer. The revised version shows significant improvement and addresses most of my previous concerns. However, a few statements remain imprecise or overly strong, and adjusting them would help ensure that the conclusions accurately reflect what can be supported by the analyses. Please see the detailed comments below.

**Major Comments:**

1. Some of the statements regarding the causality are too strong. The manuscript frequently uses expressions such as 'causally'. I understand that the main aim is to isolate the influence from the stratospheric forcing, and the authors wish to emphasize this aspect. However, it might be misleading in several contexts. For instance, the last sentence in the abstract, while it is true that the tropospheric response in these experiments originates from the stratospheric forcing, the sentence 'the observed surface response … causally forced by, stratospheric perturbations' (L18-20) is not precise. It is also unclear whether 'observed' refers to the detected model response or observational data. If the latter, the statement is too strong. I suggest rephrasing this as 'can be forced by …' or something similar. In addition, the divergent mass streamfunction can help explain the surface temperature response, but only to a certain extent. The phrase 'causally linked' (L17) might overstate the role of this diagnostic. I recommend softening the tone so that the conclusions more precisely reflect what can be inferred from the present analysis.

We agree that the mass streamfunction by the divergent meridional wind can explain only a part of the surface temperature responses. We also acknowledge that we can't demonstrate causality of the signal from observational data using these experiments. The word "causally" is removed from L17 in the revised manuscript.

However, it is not ambiguous that the surface response following SSWs can be causally forced by stratospheric perturbations, based on nudging an SSW in a model (Hitchcock and Simpson, 2014) or MiMA simulations by imposing zonally symmetric forcing in the stratosphere (White et al., 2020, 2022). In observations, the tropospheric precursors could also impact the surface conditions, and it's impossible to fully isolate the impact from the stratosphere and troposphere in the observations because they are coupled with each other. While in this study, we can fully isolate the tropospheric precursors and only address the role of the stratosphere.

2. The statement regarding the consistency with observation/reanalysis is not accurate. The manuscript currently states that the two phases correspond to clusters 4 and 5 of Kretschmer et al. (2018a). However, while the surface temperature response indeed show similarities, the stratospheric circulation does not align in the same way. In particular, wave reflection only appears in cluster 4, where cluster 5 shows a reduced upward wave propagation, but the raw Fz is still positive over Eurasia, unlike the negative raw Fz in the phase-270 experiments here. I understand that we cannot expect the idealized experiments to reproduce every observational feature. However, the comparison should be presented more carefully to avoid implying equivalence where the mechanisms differ. I suggest refining the relevant statements accordingly. In addition, repeatedly referring to cluster numbers may confuse readers unfamiliar with the cited work; it may be clearer to describe their defining characteristics when first introduced and avoid relying solely on "cluster 4/5" labels thereafter.

We agree that cluster 5 of Kretschmer et al. (2018) is not a pure composite of wave reflection events, and the stratospheric circulation in their clusters 4 and 5 is not fully consistent with phase-90 and phase-270 ensembles in our study. However, the zonal dipole characterized by positive Fz anomalies over North America and negative Fz anomalies over North Eurasia does indicate that some wave reflection events are included in cluster 5. The relevant comparison has been more carefully presented in the revised manuscript in L321-329, together with a simple description of cluster 4/5. Also shown here:

"A similar spatial pattern is also observed in the composite of cluster 4 events in Kretschmer et al. (2018a), which are characterized by raw positive Fz over Eurasia and raw negative Fz over North America. They demonstrated that such events are associated with North American cold spells via reflected upward-propagating waves over eastern Siberia by using causal effect network analysis. In the phase-270 ensemble, downward wave propagation is instead concentrated over North Eurasia, and is immediately followed by pronounced surface cooling over North Eurasia (Figure 4d). Moreover, observations of cluster 5 in Kretschmer et al. (2018a), characterized by anomalously positive Fz over North America and anomalously negative Fz over Eurasia, show a similar surface temperature response to that in the phase-270 ensemble. However, it is important to understand that cluster 5 in Kretschmer et al. (2018a) does not represent a pure composite of reflection events, as it still displays raw positive Fz over Eurasia despite the anomalously negative Fz. "

3. The alignment of timing between branch ensembles and control runs. While the authors noted that 'there is no expectation for the timing of the surface responses to match', the magnitude of response in the branch ensembles after day 13 appears more comparable to those in the control runs after day1. I understand that 'day0' represents different reference points, and strict alignment is not required. But align the timelines based on the peak zonal-wind reversal (e.g., day 0 in the control run and day 12 in the branch runs) may make the comparison more straightforward for readers. Alternatively, omitting direct cross-experiment comparisons at fixed lags, or explicitly noting their limitations, would avoid potential confusion.

Good suggestion, thanks! We have redefined day-0 in the CONTROL runs so that the day with peak easterlies is the same in both CONTROL and branch runs. It turns out that the peak

easterlies in the branch runs are on day 12 after branching (the last day with imposed forcing), and so we had to redefine day-0 in the control runs accordingly. We have revised all figures in the paper to use this new definition.

This has been clarified in the methods section in L171-174. Also shown here:

"Day 12 in control runs indicates the day with peak easterlies following the onset of SSWs, whereas day 12 in branch ensembles corresponds to January 12 of every year (the last day with imposed forcing), accompanied by peak easterlies following the onset of forced SSWs too."

**Specific Comments:**

1. L17. Should 'downward propagation events' refer instead to 'wave reflection events'?

The wave reflection could be affected by the imposed momentum torque during the forcing stage (days 1-12), while wave reflection freely responds to the zonal asymmetry of the polar vortex when the forcing is switched off (day 13 and onward). To better summarize the two periods, "downward propagation events" is used.

2. L61-63 and L408-410. Previous studies have shown that this type of stratospheric anomaly is linked to preceding tropospheric circulation (e.g., Shen et al. 2023; Tan and Bao 2020) and that similar stratospheric disturbances can lead to distinct surface response depending on the tropospheric processes involved (e.g., Shen et al. 2025). Adding a brief discussion where relevant can be helpful to strengthen the motivation for isolating the role of stratosphere.

Cited in suitable places.

3. L152. Change 'present' to 'represent'.

Corrected.

4. L203. Should be 'McIntyre' and 'Edmon et al.'.

They should be "Andrews and Mcintyre, 1976" and "Edmon Jr et al., 1980" as in the manuscript. See the following links in Google Scholar.

https://scholar.google.com/scholar?hl=en&as_sdt=0%2C5&q=Planetary+waves+in+horizontal+and+vertical+shear%3A+The+generalized+Eliassen-Palm+relation+and+the+mean+zonal+acceleration&btnG=

https://scholar.google.com/scholar?hl=en&as_sdt=0%2C5&q=Eliassen-Palm+cross+sections+for+the+troposphere&btnG=

5. L231. It is more accurate to state 'averaged over days 6 to 12'. The same applies to other similar descriptions.

Corrected.

6. Figure 3c and d. The tropospheric polar-cap height anomaly peaks almost simultaneously with the stratospheric anomaly. Could the authors clarify why this occurs?

In shorter timescales (e.g., 6-hourly), the tropospheric response lags behind the stratospheric anomalies.

7. L273-275. For the reasons described in major comment #2, I suggest reducing the emphasis on direct comparison with Kretschmer et al. (2018a), particularly for phase-270, which does not closely resemble cluster 5 beyond the surface temperature pattern.

This comparison has been removed from the revised manuscript.

8. L283-284. Here the comparison uses days 1-5, but earlier the authors note that the timings are not expected to match. As mentioned in major comment #3, aligning the timing or avoiding such direct comparisons may reduce confusion.

Your suggestion in major comment 3 was a good one - thanks! All figures have been modified to align the timing.

9. L310-312. Please specify the longitude range of the region discussed for easier interpretation.

The longitude range has been clarified in L308-312. Also shown here:

"The shading in Figure 6 shows the raw geopotential height eddy area-averaged from 55°N to 80°N, and there is a clear eastward tilt from 180°E to 300°E in the phase-90 ensemble (Figure 6c) and especially in the phase-270 ensemble (Figure 6d) with downward directed arrows from 90°E to 270°E, which also indicates the presence of wave reflection events over there (Cohen et al., 2022)."

10. L321-326. Phase-90 shares characteristics with cluster 4, but phase-270 does not resemble cluster5. Revising this statement for accuracy would be beneficial.

Good suggestion. This part has been modified in L321-329. Also shown here:

"A similar spatial pattern is also observed in the composite of cluster 4 events in Kretschmer et al. (2018a), which are characterized by raw positive Fz over Eurasia and raw negative Fz over North America. They demonstrated that such events are associated with North American cold spells via reflected upwardpropagating waves over eastern Siberia by using causal effect network analysis. In the phase-270 ensemble, downward wave propagation is instead concentrated over North Eurasia, and is immediately followed by pronounced surface cooling over North Eurasia (Figure 4d). Moreover, observations of cluster 5 in Kretschmer et al. (2018a), characterized by anomalously positive Fz over North America and anomalously negative Fz over Eurasia, show a similar surface temperature response to that in the phase-270 ensemble. However, it is important to understand that cluster 5 in Kretschmer et al. (2018a) does not

represent a pure composite of reflection events, as it still displays raw positive Fz over Eurasia despite the anomalously negative Fz. "

11. L445. Cluster 5 in Kretschmer 2018a does not show a wave reflection. This should be corrected.

Corrected in L448-449. Also shown here:

"However, it is concentrated over North Eurasia in the phase-270 ensemble (Figure 7d), consistent with the anomalous Fz pattern in cluster 5 observed by Kretschmer et al. (2018a)."

**References**

Hitchcock, P., and I. R. Simpson, 2014: The Downward Influence of Stratospheric Sudden Warmings. J. Atmos. Sci., 71, 3856–3876, https://doi.org/10.1175/JAS-D-14-0012.1

White, I. P., C. I. Garfinkel, E. P. Gerber, M. Jucker, P. Hitchcock, and J. Rao, 2020: The Generic Nature of the Tropospheric Response to Sudden Stratospheric Warmings. J. Climate, 33, 5589–5610, https://doi.org/10.1175/JCLI-D-19-0697.1

White, I. P., C. I. Garfinkel, and P. Hitchcock, 2022: On the Tropospheric Response to Transient Stratospheric Momentum Torques. J. Atmos. Sci., 79, 2041-2058, https://doi.org/10.1175/JAS-D-21-0237.1